

# Chemical Characterisation of Water-soluble Ions in Atmospheric Particulate Matter on the East Coast of Peninsular Malaysia

Naomi J. Farren[1], Rachel E. Dunmore[1], Mohammed Iqbal Mead[2], Mohd Shahrul Mohd Nadzir[3,4], Azizan Abu Samah[5], Siew-Moi Phang[5], William T. Sturges[6], Jacqueline F. Hamilton[1].

[1]Wolfson Atmospheric Chemistry Laboratories, Department of Chemistry, University of York, York, YO10 5DD, UK.
[2]Centre for Atmospheric Informatics and Emissions Technology, School of Energy, Environment and Agrifood/Environmental Technology, Cranfield University, Cranfield, UK.
[3]Centre for Tropical Climate Change System (IKLIM), Institute of Climate Change, Universiti Kebangsaan Malaysia, 43600 Bangi, Selangor, Malaysia.
[4]School of Environmental Science and Natural Resources, Faculty of Science and Technology, Universiti Kebangsaan Malaysia, 43600 Bangi, Selangor Darul Ehsan, Malaysia.
[5]Institute of Ocean and Earth Sciences, University of Malaya, Kuala Lumpur, Malaysia.
[6]Centre for Ocean and Atmospheric Sciences, School of Environmental Sciences, University of East Anglia, Norwich, UK.

*Correspondence to*: Jacqueline F. Hamilton (jacqui.hamilton@york.ac.uk)

**Abstract.**

Air quality on the east coast of Peninsular Malaysia is influenced by local anthropogenic and biogenic emissions, as well as marine air masses from the South China Sea and aged emissions transported from highly polluted East Asian regions during the winter monsoon season. An atmospheric observation tower has been constructed on this coastline at the Bachok Marine and Atmospheric Research Station. Daily $PM_{2.5}$ samples were collected from the top of the observation tower over a 3-week period, and ion chromatography was used to make time-resolved measurements of major atmospheric ions present in aerosol. $SO_4^{2-}$ was found to be the most dominant ion present, and on average made up 66% of the total ion content. Predictions of aerosol pH were made using the ISOROPPIA-II thermodynamic model and it was estimated that the aerosol was highly acidic, with pH values ranging from -0.97 to 1.12. A clear difference in aerosol composition was found between continental air masses originating from industrialised regions of East Asia and marine air masses predominantly influenced by the South China Sea. For example, elevated $SO_4^{2-}$ concentrations and increased $Cl^-$ depletion was observed when continental air masses that had passed over highly industrialised regions of East Asia arrived at the measurement site. Correlation analyses of the ionic species and assessment of ratios between different ions provided an insight into common sources and formation pathways of key atmospheric ions, such as $SO_4^{2-}$, $NH_4^+$ and $C_2O_4^{2-}$. To our knowledge, time-resolved measurements of water-soluble ions in $PM_{2.5}$ are virtually non-existent in rural locations on the east coast of Peninsular Malaysia; overall this dataset contributes towards a better understanding of atmospheric composition in the Maritime Continent, a region of the tropics that is vulnerable to the effects of poor air quality, largely as a result of rapid industrialisation in East Asia.





## 1 Introduction

The tropical Maritime Continent, a region in Southeast Asia between 10° S – 20 °N and 90° - 150° E, is a complex distribution of islands and peninsulas, and incorporates countries such as Malaysia, Indonesia, the Philippines and Papua New Guinea (Neale and Slingo, 2003). It lies within a tropical warm pool that extends eastwards from the Indian Ocean to the Western Pacific, and is home to some of the warmest ocean temperatures in the world. Tropical regions such as the Maritime Continent are of central importance for the chemistry-climate system (Carpenter et al., 2010). For example, high photochemical activity

in these regions means that global atmospheric lifetimes of key atmospheric species, such as methane and ozone, are controlled by destruction rates in the tropics (Lawrence et al., 2001; Bloss et al., 2005). In terms of ocean productivity, the observed decrease in primary productivity in low-latitude oceans has been linked to a reduced availability of nutrients for phytoplankton growth, caused by changes in upper-ocean temperature and stratification (Behrenfeld et al., 2006). Furthermore, the wind circulation system in the Maritime Continent is influenced by seasonal Asian monsoons, which are controlled by the natural

oscillation of the intertropical convergence zone (ITCZ). During the northern hemisphere winter, a large anticyclone forms over Siberia each year, creating strong north-easterly monsoon winds in the South China Sea (Northeast Monsoon). These strong north-easterlies can transport air masses from rapidly developing East Asian countries (*e.g.* China, Japan, Taiwan, Vietnam, North and South Korea) across the South China Sea to the Maritime Continent. In addition, cold surge events occur regularly throughout the winter monsoon season and last several days. Cold surges occur as a result of a south-easterly

movement of the anticyclone, and are characterised by cold air masses over Southern China and strengthening of the north-easterly monsoon winds in the South China Sea (Zhang et al., 1997). The transport of pollution from East Asia to the tropics during the monsoon season, particularly during cold surges, means that rural areas such as the east coast of Peninsular Malaysia are potentially at an elevated risk of the detrimental effects of poor air quality.

Tropical regions are highly important for atmospheric research, and whilst long-term atmospheric observations exist (Robinson et al., 2014; Pyle et al., 2011), there are fewer measurements than in the mid and high-latitudes. The Bachok Marine and Atmospheric Research Station (6.00892° N, 102.42504° E) has been set up on the east coast of Peninsular Malaysia and is ideally located for studying the outflow of these highly industrialised regions, and for investigating the interaction with cleaner air in the Southern hemisphere. The research station forms part of the Institute of Ocean and Earth Sciences at the University

of Malaya (UM), and is located approximately 30 km away from Kota Bharu. An atmospheric observation tower facing the South China Sea has been constructed at the research station; this has been built for the specific purpose of monitoring long-range transported pollution, air sea exchange and coastal meteorology. The research station is working towards designation as a regional Global Atmospheric Watch (GAW) centre, which will be a valuable addition to the network of other global and regional GAW sites in the Maritime Continent, as shown in Fig. 1 (gawsis.ch, 2017).


**Figure 1: Location of global (green pins) and regional (blue pins) GAW sites in the Maritime Continent; Danum Valley in Malaysia (DMV), Bukit Kototabang in Indonesia (BKT), Manila in Philippines (MNI), Songkhla in Thailand (SKH), Tanah Rata in Malaysia**





**(TAR), Petaling Jaya in Malaysia (PJM) and Singapore (SIN) (gawsis.ch, 2017). The red pin shows the location of the Bachok Marine and Atmospheric Research Station. Map created using google maps (google.com, 2017).**


In January and February 2014, an instrument demonstration campaign was carried out to assess the capabilities of the new research station. This was funded by the Natural Environment Research Council (NERC) and UM and involved several UK universities, as well as the National Centre for Atmospheric Science (NCAS), UM and the Malaysia Meteorological Department (MMD). As part of this study, Dunmore et al. used a specialised multi-dimensional gas chromatography technique to accurately measure atmospheric mixing ratios of $C_5$-$C_{13}$ VOC species with a wide range of functionalities (Dunmore et al., 2016). Furthermore, Dominick et al. characterised the particulate matter in Bachok by studying the influence of north-easterly winds on the patterns of particle mass and particle number concentration size distributions (Dominick et al., 2015). Both studies highlighted the fact that the site is influenced by a mixture of local anthropogenic and biogenic emissions, clean marine air masses, and aged emissions transported from East Asia.


To extend upon these studies, it is important to investigate atmospheric aerosol composition in the Bachok region. A better understanding of aerosol chemical composition is essential as aerosols play an important role in atmospheric processes and climate change. For example, aerosols can modify the global radiation budget both directly, by scattering and absorbing solar radiation, and indirectly, by altering cloud properties and lifetime (Charlson et al., 1991). The strength of these direct and indirect effects depends partly on the particle concentration and size distribution, but also on the chemical composition.

There are a limited number of studies that focus on particulate matter composition on the east coast of Peninsular Malaysia, and to our knowledge the composition of ionic species has not been determined at any rural locations along this coastline. For example, Tahir et al. studied the composition of major elements and water-soluble ionic species in $PM_{2.5}$ and $PM_{10}$ samples on the east coast of Peninsular Malaysia, but the samples were collected at an urban coastal city, Kuala Terengganu (Tahir et al., 2013). This study used principal component analysis to determine the main sources of both fine and coarse particles, which were found to be soil dust, marine aerosol, vehicle exhaust, secondary aerosol, road dust and biomass burning. In addition, Ismail et al. studied $PM_{10}$ concentrations in three major cities (Kota Bahru, Kuala Terengganu and Kuantan) on the east coast of Peninsular Malaysia between 2006 and 2012 (Ismail et al., 2016). The study showed that during the Northeast Monsoon, the air arriving at the sites had originated from China and the Philippines and travelled over the South China Sea. During the Southwest Monsoon, the air came from Indonesia *via* the Straits of Malacca. Over the 6-year period, it was found that the atmospheric $PM_{10}$ mass was directly proportional to the rate of urbanization in each of the three cities.

In this study, measurements of water-soluble ions in atmospheric aerosol at a rural coastal location on the east coast of Peninsular Malaysia are presented. Analysis of temporal variation of different ionic species has been carried out, and backward air mass trajectories have been used to determine the influence of air mass origin on aerosol composition. Correlation analyses



of the ionic species and assessment of ratios between different ions has provided an insight into common sources and formation pathways of key atmospheric ions.

## 2 Experimental

### 2.1 Eluents and analytical standards

Ultrapure milli-Q water (18 MΩ cm$^{-1}$) from an ELGA LabWater purification system was used to prepare all the required eluents and analytical standards. A 20 mM solution of methanesulfonic acid was used as the eluent for cation exchange chromatography and for anion exchange chromatography, a solution of 8 mM $Na_2CO_3$/1 mM $NaHCO_3$ was prepared. Using a variety of salts and organic acids, individual analytical standards containing 500 ppm of each target ion ($Cl^-$, $NO_2^-$, $NO_3^-$, $PO_4^{3-}$, $SO_4^{2-}$, $CH_3SO_3^-$, $C_2O_4^{2-}$, $Na^+$, $NH_4^+$, $K^+$, $Mg^{2+}$ and $Ca^{2+}$) were prepared in milli-Q water. The salts and organic acids were purchased from either Sigma-Aldrich Ltd. (Dorset, UK) or Fisher Scientific Ltd. (Loughborough, UK).

### 2.2 Method validation

Recovery tests were performed by spiking 5.7 cm$^2$ of quartz fibre filters (Whatman, Maidstone, UK) with 1 µg of each target ion (20 µL of a 50 ppm mixed ion solution). Prior to spiking, the filters were prebaked at 550 °C for 6 hours and wrapped in aluminium foil and stored at -18 °C until required. The spiked filters were dissolved in 2 mL of milli-Q water and sonicated for 30 min at room temperature. The extract was filtered using a Millex-GP 33 mm diameter hydrophilic syringe filter with a pore size of 0.22 µm (Millipore UK Limited, Watford, UK) and made up to a final volume of 2.5 mL. Procedural blanks were also carried out using quartz fibre filters (5.7 cm$^2$) and blank subtractions were applied to any target ions found in detectable amounts.

### 2.3 Sample collection and extraction

Thirty PM$_{2.5}$ samples were collected at the Bachok Marine and Atmospheric Research Station (6.00892° N, 102.42504° E) between 18-01-2014 and 06-02-2014. The samples were collected at the top of an atmospheric observation tower (18 m height) using a high volume air sampler (Ecotech HiVol 3000, Victoria, Australia) operating at 1.13 m$^3$ min$^{-1}$ over 24 h sampling intervals. The tower is located on the coastline of the South China Sea and is within 100 m of the shore. A 3-day intensive measurement period was in operation between midday (local time) on 30-01-2014 and midday on 02-02-2014, in which filters were collected every 4-8 hours. The quartz fibre filters (20.3 × 25.4 cm) supplied by Whatman (Maidstone, UK) were prebaked at 550 °C for a minimum of 12 h prior to sample collection. After sample collection, the filters were wrapped in aluminium foil and stored at -18 °C until analysis. To prepare the samples for analysis, 5.7 cm$^2$ of each sample was dissolved in 2 mL milli-Q water and sonicated for 30 min at room temperature. The extract was filtered using a Millex-GP 33 mm diameter hydrophilic syringe filter with a pore size of 0.22 µm (Millipore UK Limited, Watford, UK) and made up to a final volume of 2.5 mL.



## 2.4 Chromatographic analysis

Chromatographic analysis was carried out using a Thermo Scientific Dionex ICS-1100 ion chromatography system equipped
with an AS-DV autosampler. The column configuration used for anion exchange consisted of an IonPac AG14A guard column
($4 \times 50$ mm) and an IonPac AS14A analytical column ($4 \times 250$ mm). Cation exchange chromatography was performed using
an IonPac CG12A guard column ($4 \times 50$ mm) and an IonPac CS12A analytical column ($4 \times 250$ mm). ASRS 300 and
CSRS 300 self-regenerating suppressors (4 mm) were used for anion and cation exchange respectively. All columns and
suppressors were supplied from Thermo Scientific Dionex. The run times for the anion and cation separations were 18 and
15 min respectively. The suppressor current was 45 mA for anion exchange mode, and 59 mA for cation exchange mode. For
all separations, the instrument was operated in isocratic mode at a flow rate of 1 mL min$^{-1}$ and a column oven temperature of
30 °C. The injection volume was 100 µL and the data collection rate was 5 Hz. The system relied on a DS6 heated conductivity
cell for ion detection and all data was analysed using Thermo Scientific Chromeleon 7.1 Chromatography Data System
software.

## 2.5 Additional measurements

Individual volatile organic compounds (VOCs) were measured using a combined heart-cut and comprehensive
two-dimensional gas chromatography system (GC-GC×GC); a detailed description of the instrument design is provided in a
separate study (Dunmore et al., 2016). Measurements of NO and NO$_2$ were performed using a two channel TE42i commercial
gas analyser (Thermo Scientific, MA, USA), and SO$_2$ measurements were made using a Thermo Scientific 43i SO$_2$ analyser.
Meteorological data was accessed from the Integrated Surface Database (NOAA, 2003). The data selected was from the nearest
meteorological station, approximately 23 km away from the Bachok measurement site at the Sultan Ismail Petra Airport
(6.17208° N, 102.29288° E). Hourly measurements of wind direction, wind speed, air temperature, dew point, atmospheric
pressure and relative humidity were obtained. 10-day backward air mass trajectories arriving at the sampling site were run
every 3 hours throughout the entire measurement period. A receptor height of 10 m was chosen to represent the measurements
made on the sampling tower. The trajectories were computed using the Hybrid Single-Particle Lagrangian Integrated
Trajectory (HYSPLIT) model (Stein et al., 2015; Draxler, 1999; Draxler and Hess, 1998, 1997), and the data was analysed
using the openair package in RStudio (Carslaw and Ropkins, 2012; Carslaw, 2015).

## 2.6 ISORROPIA-II model

Predictions of aerosol pH were made using the ISORROPIA-II thermodynamic equilibrium model (Fountoukis and Nenes,
2007). Calculations were made in 'reverse mode', in which known quantities are temperature, relative humidity, and particle
phase concentrations of NH$_4^+$, SO$_4^{2-}$, Na$^+$, Cl$^-$, NO$_3^-$, Ca$^{2+}$, K$^+$ and Mg$^{2+}$. The aerosol was assumed to be thermodynamically



stable *i.e.* the aerosol can exist as both solid and liquid, and salts are able to precipitate if the aqueous phase becomes saturated with respect to them.

## 3 Results and discussion

### 3.1 Method validation

Using isocratic elution methods for both cation and anion exchange chromatography, the target ions were successfully separated. The recovery of the target ions from the filter papers ranged from 74.5% to 98.2% for the target anions, and 78.3% to 87.3% for the target cations, with the exception of $Ca^{2+}$ for which a recovery level of 123.3% was calculated. Recovery tests were carried out in triplicate and $\%RSD_{rec}$ remained below 8% for all the ions. The recovery of $Ca^{2+}$ should not have exceeded 100% and the result may be attributed to inconsistencies in the amount of $Ca^{2+}$ present on the blank filter, or due to $Ca^{2+}$ contamination during sample collection or storage. 100% recovery was assumed for $Ca^{2+}$ during the data analysis process. All the reported ion concentrations in this study have been corrected for procedural blanks. Further details of the individual recovery levels and associated errors, as well as the blank contribution of each ion can be found in the Supplement (Table S1). The main instrumental parameters of the IC system were evaluated and are also detailed in the Supplement (Table S2). Instrumental limits of detection (LODs) and limits of quantification (LOQs) were calculated according to the EPA protocol 40 CFR 136; multiplying the standard deviation (N = 10, 5 ng for cations, 25 ng for anions) by the Student t-value (N = 10, 95% confidence interval) gave the LOD, and multiplying the standard deviation by 10 gave the LOQ (EPA, 2017; Ripp, 1996). For anion exchange chromatography, LODs and LOQs were in the range 5.5 – 21.0 ng and 25.3 – 144.2 ng respectively. For cation exchange chromatography, the LODs ranged from 0.5 to 2.1 ng and the LOQs ranged from 2.5 to 6.1 ng. On average, the instrument precision ($\%RSD_{ins}$, n = 10) was 4.2% for the target cations and 12.8% for the target anions. Total errors were estimated by combining errors with the instrument and the recovery process and remained below 15.4% for all ions except $NO_3^-$ (22.6%). In summary, the IC method proved to be a reliable technique, allowing for water-soluble ions in atmospheric aerosol to be quantified accurately.

### 3.2 Bachok demonstration campaign

The filter samples were collected at the Bachok atmospheric observation tower. The Bachok district, located in the state of Kelantan, is a rural area and the primary economic activity comes from tobacco and kenaf plantations. Other agrarian activities in the wider Kelantan region include the production of rice and rubber, as well as additional economic activities such as livestock rearing and fishing. Figure 2 shows the 10-day backward air mass trajectories arriving at the measurement site during the demonstration campaign. As expected during the winter months, the strong anticyclone system known as the Siberian High led to the arrival of north-easterly onshore winds along the east coast of Peninsular Malaysia. Some of the air masses experienced a significant continental influence from highly industrialised countries such as China, Japan, Taiwan and North



and South Korea, whilst other air masses had a stronger marine influence from both the East China Sea and the South China Sea.

**Figure 2: 10-day HYSPLIT backward air mass trajectories centred on the Bachok Marine and Atmospheric Research Station between 18-01-2014 and 07-02-2014. Plot constructed using the openair package in RStudio (Carslaw and Ropkins, 2012; Carslaw, 2015).**

Although there was no reliable meteorological data recorded at the measurement site during the demonstration campaign, data
from a nearby meteorological station was available. The station is located approximately 23 km away at the Sultan Ismail Petra airport in Kota Bharu (6.17298° N, 102.29288° E), as shown in Fig. S1 (Supplement). Whilst the meteorological data from the airport will not be exactly representative of the measurement site, the patterns in wind direction are consistent with observations made by the field scientists during the campaign. In addition, during a study of the influence of Northeast Monsoon cold surges on air quality in Southeast Asia, Ashfold et al. used meteorological data from three locations that they believed to lie in the
path of cold surges during the Northeast Monsoon, offering the best possibility of observing a cold surge influence on air pollution (Ashfold et al., 2017). These sites were in Kota Bharu (102.247° E, 6.141° N), Kuala Terengganu (103.118° E, 5.308° N) and Kemaman (103.428° E, 4.271° N); this provides further confirmation that data from the Sultan Ismail Petra airport site is suitable for evaluating broad scale transport at the Bachok measurement site. Fig. S2 in the Supplement shows average hourly wind speed and wind direction conditions across the entire duration of the measurement campaign (18-01-2014
to 06-02-2014). On most days, gentle south westerlies from the land (Peninsular Malaysia) were observed in the early hours of the morning, through to around 11 am. At this stage, a dramatic shift in wind direction occurred, and until around 19:00 a strong onshore breeze from the north east (South China Sea) was observed. From 20:00 to the early hours of each morning, a calmer sea breeze predominantly from the east was seen. Hourly VOC measurements were conducted at the measurement site, and the development of a sea breeze at approximately 11:00 dramatically influenced the diurnal profiles of the measured
species (Fig. S3, Supplement). In the morning, when the air being sampled was coming over the land, high levels of NO, $NO_2$ and anthropogenic VOCs such as toluene and $C_{10}$ aliphatics were observed; the main source of these species was local burning of waste (Dunmore et al., 2016). When the sea breeze developed the concentration of these species dropped significantly.

### 3.3 Composition of water-soluble ions in atmospheric aerosol

### 3.3.1 Aerosol composition and determination of non-sea salt and sea salt components

The total concentration of measured water-soluble ions in the $PM_{2.5}$ during the campaign ranged from 8.06 to 27.0 µg m⁻³, with an average concentration of 16.2 µg m⁻³. A study of particle mass and number concentration at the Bachok measurement site was carried out by Dominick et al.; average $PM_{2.5}$ and $PM_{10}$ concentrations of 30 µg m⁻³ and 31 µg m⁻³ were observed respectively between 09-01-2014 and 23-03-2014 (Dominick et al., 2015). This data suggests that on average, the measured water-soluble ions in this study made up approximately half of the total $PM_{2.5}$, and it is likely that the remainder was comprised



primarily of organic aerosol. During the measurement campaign, PM levels regularly exceeded the WHO PM$_{2.5}$ guidelines for

the 24-hour mean (25 µg m$^{-3}$) (WHO, 2006). Table 1 shows the mean and maximum water-soluble ion concentrations measured

throughout the campaign, and Figures 3 and 4 show time series for all the ions measured in the aerosol.

**Table 1: Mean and maximum ion concentrations measured throughout the measurement period. The average % mass contribution**

**of each ion to the total measured ions is included, as well as the % of samples in which each target ion is found (%Qt).**

**Figure 3: Time series of SO$_4^{2-}$, NH$_4^+$, Na$^+$, Cl$^-$, NO$_3^-$ and NO$_2^-$ concentration (µg m$^{-3}$) during the Bachok demonstration campaign**

**(18-01-2014 to 07-02-2014). Yellow shaded areas represent the time between sunrise and sunset (local).**

**Figure 4: Time series of PO$_4^{3-}$, Ca$^{2+}$, Mg$^{2+}$, K$^+$, C$_2$O$_4^{2-}$ and CH$_3$SO$_3^-$ concentration (µg m$^{-3}$) during the Bachok demonstration**

**campaign (18-01-2014 to 07-02-2014). Yellow shaded areas represent the time between sunrise and sunset (local).**

As the composition of the water-soluble ions present in aerosol collected at the Bachok site was influenced by both marine

and continental sources, it is useful to make an estimation of non-sea salt (*nss*) and sea salt (*ss*) components, using

Eq. (1) – Eq. (4). Total Na$^+$ and Ca$^{2+}$ concentrations have been measured in this study, and the mean Na$^+$/Ca$^{2+}$ ratio in the crust

and mean Ca$^{2+}$/Na$^+$ ratio in seawater have been estimated as 1.78 *w/w* and 0.038 *w/w* respectively (Bowen, 1979). Therefore

is it possible to solve Eq. (1) – Eq. (4) simultaneously for *ss*Na$^+$, *nss*Na$^+$, *ss*Ca$^{2+}$ and *nss*Ca$^{2+}$ (Boreddy and Kawamura, 2015).

Furthermore, the resulting estimate of *ss*Na$^+$, which can be used as a sea spray marker, can also be used to predict the

contribution of *nss*SO$_4^{2-}$ and *nss*K$^+$ in the aerosol, as shown in Eq. (5) and Eq. (6) respectively.


$$ssNa^+ = Na^+ - nssNa^+ \tag{1}$$

$$nssNa^+ = nssCa^{2+} \cdot \left(\frac{Na^+}{Ca^{2+}}\right)_{crust} \tag{2}$$

$$nssCa^{2+} = Ca^{2+} - ssCa^{2+} \tag{3}$$

$$ssCa^{2+} = ssNa^+ \cdot \left(\frac{Ca^{2+}}{Na^+}\right)_{sea\ water} \tag{4}$$

$$nssSO_4^{2-} = SO_4^{2-} - 0.253 \cdot ssNa^+ \tag{5}$$

$$nssK^+ = K^+ - 0.037 \cdot ssNa^+ \tag{6}$$

Figure 5 shows a series of pie charts to summarise the average mass composition of water-soluble ions in atmospheric aerosol,

and the distribution of non-sea salt and sea salt components. The results show that the water-soluble ion fraction of the aerosol

is dominated by SO$_4^{2-}$, which on average made up 65.6% of the total ion content by mass. NH$_4^+$ and NO$_3^-$ concentrations were

significantly lower, with mean concentrations of 1.69 and 0.61 µg m$^{-3}$ respectively. Na$^+$ and Cl$^-$ made up 11.1% of the total

ion content, and 75% of the measured Na$^+$ was attributed to *ss*Na$^+$. The average concentrations of *nss*K$^+$ and *nss*Ca$^{2+}$ were 0.30

and 0.09 µg m$^{-3}$ respectively; these ions can be used as tracers for biomass burning (*nss*K$^+$) and atmospheric dust (*nss*Ca$^{2+}$).



$NO_2^-$ and $CH_3SO_3^-$ were the least abundant ions, with average concentrations of 0.05 and 0.08 µg m$^{-3}$. The two ions were only observed in a subset of samples; $CH_3SO_3^-$ was quantified in 67% of samples and quantification of $NO_2^-$ was only achieved in 23% of samples.

**Figure 5: Pie chart to show the average composition of water-soluble ions in aerosol collected at the Bachok research station (upper panel) and pie charts to show the percentage of non-sea salt and sea salt fractions of Na$^+$, SO$_4^{2-}$, K$^+$, Ca$^{2+}$ (lower panel, left to right).**

**3.3.2 Sources and formation of sulfate (SO$_4^{2-}$)**

The average $SO_4^{2-}$ concentration during the measurement campaign was 10.7 µg m$^{-3}$, with a maximum concentration of 20.8 µg m$^{-3}$ recorded. The formation of $SO_4^{2-}$ in the particle phase occurs when emitted $SO_2$ is oxidised by OH in the gas phase, or by $O_3$ or $H_2O_2$ in the aqueous phase (Fisher et al., 2011). The most dominant anthropogenic sources of $SO_2$ include fuel and industrial emissions, as well as open biomass burning. Natural sources of $SO_2$ arise from both volcanic activity and the 270 oxidation of biogenic dimethyl sulphide (DMS) (Fisher et al., 2011).

By using $ss$Na$^+$ as a sea spray marker to determine non-sea salt and sea salt components of the aerosol, it was found that on average 96% of the measured $SO_4^{2-}$ was $nss$SO$_4^{2-}$, and only 4% of the $SO_4^{2-}$ was from sea salt. As a potential biogenic source of $nss$SO$_4^{2-}$ is DMS emissions from marine biota, and the main atmospheric source of MSA is the oxidation of DMS, it is 275 possible to use the MSA$^-$/$nss$SO$_4^{2-}$ ratio as a tracer to assess the contribution of biogenic sources to $nss$SO$_4^{2-}$ in the atmosphere (Legrand and Pasteur, 1998). In this study, MSA$^-$ concentrations ranged from 0.02 to 0.22 µg m$^{-3}$, with an average concentration of 0.08 µg m$^{-3}$. As a result, the MSA$^-$/$nss$SO$_4^{2-}$ ratio ranged from 2.5×10$^{-4}$ to 2.3×10$^{-3}$, with an average value of 7.8×10$^{-4}$. These values are very low compared to MSA$^-$/$nss$SO$_4^{2-}$ ratios recorded at remote sites; for example, a MSA$^-$/$nss$SO$_4^{2-}$ mean mass ratio of 0.07 has been measured on Fanning Island and American Samoa (Savoie and Prospero, 1989). The lower 280 ratios recorded in Bachok suggest the majority of $nss$SO$_4^{2-}$ at the site originates from anthropogenic sources.

As previously discussed, a study by Dunmore et al. revealed that levels of NO$_x$ and anthropogenic VOCs at the Bachok measurement site were significantly higher when the air being sampled had passed over nearby land, and dropped significantly at around 11 am when a sea breeze developed (Dunmore et al., 2016). This indicated that air quality in Bachok is influenced 285 by local sources of pollution, such as vehicle emissions and burning domestic waste. A pollution rose is shown in Fig. 6 to show the relationship between gaseous $SO_2$ and wind direction. $SO_2$ concentrations below 5 ppb have been excluded in order to investigate the wind conditions when the spikes in $SO_2$ concentration occur in more detail. The majority of higher $SO_2$ events were observed in calm conditions when the air arriving at the site had passed over land to the south west of Bachok; this provides further evidence that the site is influenced by local sources of pollution.




**Figure 6: Pollution rose to show the relationship between wind direction and SO₂ concentration (≥ 5 ppb) at the Bachok measurement site. Plot constructed using the openair package in RStudio (Carslaw and Ropkins, 2012; Carslaw, 2015).**

At the Bachok measurement site, no obvious relationship was observed between $SO_2$ and particulate $SO_4^{2-}$ concentration, or

between $SO_4^{2-}$ concentration and wind direction. This is likely to be because it takes time for $SO_2$ to oxidise to $SO_4^{2-}$, and that the $SO_4^{2-}$ fraction of the aerosol is more heavily influenced by long-range transport of aged emissions from East Asia. To investigate this further, the backward air mass trajectories were coloured by the concentration of $SO_4^{2-}$, as shown in Fig. 7 (upper panel). The plot clearly shows that the $SO_4^{2-}$ content of the aerosol is highest (*ca.* 14 – 20 µg m⁻³) when the site is influenced by continental air masses from regions of  East Asia, and lowest when the air masses have a more significant marine

influence. With this information in mind, it is useful to perform cluster analysis on the back trajectories; this type of analysis groups air masses of similar origin together, which provides more information on pollutant species with similar chemical histories. Figure 7 (lower panel) shows the 10-cluster solution to back trajectories calculated for the Bachok site during the measurement campaign.

**Figure 7: Upper panel shows 10-day HYSPLIT backward air mass trajectories centred on the Bachok research station between 18-01-2014 and 07-02-2014. The back trajectories are coloured by the concentration of SO₄²⁻ (µg m⁻³). Lower panel shows the 10-cluster solution to backward air mass trajectories centred on the Bachok research station during the same time period. The clusters are coloured by the average concentration of SO₄²⁻ (µg m⁻³). Plot constructed using the openair package in RStudio (Carslaw and Ropkins, 2012; Carslaw, 2015).**


The air masses associated with the highest $SO_4^{2-}$ content are represented by clusters 4 and 10 shown in the lower panel of Fig. 7; average $SO_4^{2-}$ concentrations for these two clusters were 20.4 and 18.1 µg m⁻³ respectively. Cluster 4 contained air masses that had passed over several highly industrialised regions en route to Bachok, including cities such as Zhanjiang (China) and Ho Chi Minh City (Vietnam). Air masses in cluster 10 passed over the megacity of Manila in the Philippines, but may

have slightly lower $SO_4^{2-}$ levels due to the height of the back trajectories; the average height (above sea level) of the back trajectories in cluster 4 was 181 m over the 10-day period, whilst the average height for the trajectories in cluster 10 was 1575 m. Average $SO_4^{2-}$ concentrations for clusters 1, 3 and 5 were 11.5, 14.1 and 10.5 µg m⁻³ respectively. These air masses experienced some continental influence from the east coast of China but mainly passed over the South China Sea before arriving at Bachok. Air masses with the lowest average $SO_4^{2-}$ content (between 6.6 and 8.7 µg m⁻³) were associated with

clusters 2 and 6 – 9. Most of these air masses originated from the East China Sea and South China Sea and did not undergo any significant continental influence.

Similar observations have been reported in a separate study by Oram et al., in which chlorine-containing very short-lived substances (Cl-VSLSs) were measured at the Bachok research station during the same winter monsoon season in late January/



early February 2014 (Oram et al., 2017). A 7-day pollution or cold-surge event was reported between 19-01-2014 and 26-01-2014, when significantly enhanced concentrations of Cl-VSLSs were observed. During this pollution episode, the measured samples were heavily impacted by emissions from the East Asian mainland, whilst this influence was less significant during the cleaner, non-polluted periods. In fact, the total median concentration of the four measured Cl-VSLSs was 546 ppt between 20 and 26 Jan, and 243 ppt during the less polluted period (27-01-2014 to 05-02-2014). In this study, the mean $SO_4^{2-}$

concentrations in these two periods were 14.9 and 8.8 µg m$^{-3}$ respectively. Furthermore, Oram et al. noted that even after the cold surge event, the levels of Cl-VSLSs were still significantly higher than expected, indicating that this region of the South China Sea is widely impacted by emissions from East Asia. The widespread influence from industrial emissions on a regional scale is further evidenced in this study. For example, whilst air masses arriving at Bachok from highly industrialised regions contained higher $SO_4^{2-}$ levels, the $SO_4^{2-}$ concentration remained above 5 µg m$^{-3}$ throughout the entire measurement period. To

further investigate the frequency and duration of the pollution or cold surge events, Oram et al. performed a NAME trajectory analysis using carbon monoxide (CO) as a tracer of industrial emissions from regions north of 20° N for 6 winter seasons (2009/2010 – 2014/2015) (Oram et al., 2017). A strong correlation between CO and $CH_2Cl_2$ (a measured Cl-VSLS) was observed during the pollution episode in late January 2014. Analysis of CO time series over the 6 winter seasons revealed that cold surge events are likely to be repeated regularly each winter, demonstrating that pollution rapidly undergoes long-range

transport across the South China Sea on a regular basis during the Northeast Monsoon.

### 3.3.3 Correlation of $SO_4^{2-}$ with $NH_4^+$ and implications for aerosol acidity

Ammonium ($NH_4^+$) was the second most abundant ion in the aerosol; on average it made up 10.4% of the total ion content, and mean and maximum concentrations were 1.69 and 4.73 µg m$^{-3}$ respectively. Strong positive correlation between $SO_4^{2-}$ and $NH_4^+$ was observed (R = 0.76, p < 0.001). A similar observation was reported by Keywood et al. during an investigation of the

sources of particles contributing to haze in the Klang Valley, Malaysia (Keywood et al., 2003). The strong relationship between these species is due to neutralisation of $SO_4^{2-}$ by $NH_4^+$. It is likely that $NH_3$ emissions in the rural Bachok region come predominantly from agricultural practices such as animal husbandry, fertilizer use and agricultural waste burning. The measurement site is possibly influenced by other key $NH_3$ sources included in the Emissions Database for Global Atmospheric Research (EDGAR), such as direct soil emissions and road transport, but it is difficult to ascertain which sources are most

dominant as the database does not provide local/ regional scale $NH_3$ emissions data for Malaysia (edgar.eu, 2016).

The uptake of $SO_4^{2-}$ is preferential to the uptake of $NO_3^-$ because sulfuric acid has a lower vapour pressure than nitric acid, and aqueous or solid $(NH_4)_2SO_4$ is the preferred form of sulfate (Seinfeld and Pandis, 2006). The average $NH_4^+/SO_4^{2-}$ molar ratio was 0.81, which indicated that there was insufficient gaseous $NH_3$ in the atmosphere to neutralise $SO_4^{2-}$. Although

measurements of the total amounts of ammonia and sulfate in the gas, aqueous and solid phase would provide a better prediction of the aerosol acidity, the results presented in this study indicated that an ammonia-poor regime exists, and that the aerosol is likely to be acidic. In these scenarios, the $NH_3$ partial pressure is low, and therefore the $NH_3$-$HNO_3$ partial pressure





product is also low, meaning that the concentrations of ammonium nitrate are low or zero (Seinfeld and Pandis, 2006). This hypothesis can be supported by the fact that $NO_3^-$ concentrations in this study were very low, ranging from 0.005 to 1.52 µg m$^{-3}$.

As a result, $NO_3^-$ made up a significantly smaller fraction of the total ion content compared to $SO_4^{2-}$; average percentage mass composition of the total ion content was 3.8% for $NO_3^-$ and 65.6% for $SO_4^{2-}$.

To estimate proton loading in atmospheric particles, the strong acidity approach can be used, as shown in Eq. (7). This approach assumes that any deficit in measured cation charge compared to measured anion charge can be attributed to H$^+$. Total anion

and total cation equivalents can be estimated using Eq. (8) and Eq. (9).

$$strong\ acidity\ (\mu eq.\,m^{-3}) = \sum anion\ equivs.\ (\mu eq.\,m^{-3}) - \sum cation\ equivs.\ (\mu eq.\,m^{-3}) \tag{7}$$

$$\sum anion\ equivs.\ (\mu eq.\,m^{-3}) = \frac{SO_4^{2-}}{48} + \frac{NO_3^-}{62} + \frac{Cl^-}{35.5} + \frac{PO_4^{3-}}{31.6} + \frac{C_2O_4^{2-}}{44} + \frac{NO_2^-}{46} + \frac{CH_3SO_3^-}{95} \tag{8}$$

$$\sum cation\ equivs.\ (\mu eq.\,m^{-3}) = \frac{Na^+}{23} + \frac{NH_4^+}{18} + \frac{K^+}{39} + \frac{Mg^{2+}}{12} + \frac{Ca^{2+}}{20} \tag{9}$$


As shown in Fig. 8, strong acidity values ranged from 0.03 to 0.19 µeq. m$^{-3}$, with an average value of 0.11 µeq. m$^{-3}$. The positive strong acidity values provide an initial indication that the aerosol is acidic and allows an estimate of the proton loading to be made (average H$^+$ = 0.11 µg m$^{-3}$).

**Figure 8: Particle strong acidity and associated error predictions for the aerosol collected during the Bachok measurement campaign. Yellow shaded areas represent the time between sunrise and sunset (local).**

To obtain error bars for the strong acidity predictions (Fig. 8), H$^+_{max}$ and H$^+_{min}$ were calculated according to Eq. (10) and Eq. (11) respectively. For H$^+_{max}$ the anions were adjusted up to within their uncertainties (*i.e.* +%RSD$_{tot}$) and the cations were

adjusted down to within their uncertainties (*i.e.* -%RSD$_{tot}$). For H$^+_{min}$ the anions were adjusted down and the cations were adjusted up (Murphy et al., 2017). %RSD$_{tot}$ was estimated by combining the error of the recovery process for each ion (%RSD$_{rec}$, Table S1) and the error of the instrument for each ion (%RSD$_{ins}$, Table S2).

$$H_{max}^+ = \sum max.\ anion\ equivalents - \sum min.\ cation\ equivalents \tag{10}$$

$$H_{min}^+ = \sum min.\ anion\ equivalents - \sum max.\ cation\ equivalents \tag{11}$$

In most cases, H$^+_{min}$ remains above zero. However, between 02-02-2014 and 06-02-2014, slightly negative H$^+_{min}$ values between -1×10$^{-3}$ and -2×10$^{-2}$ µeq. m$^{-3}$ were calculated, which are physically implausible (Murphy et al., 2017). These results highlight the possible sources of error associated with the strong acidity approach for estimating aerosol acidity. For example,

Hennigan et al. report that organic acids (which are mostly excluded from this study, except for MSA and oxalic acid) can





have an important influence on aerosol acidity, especially at relatively low acidities where organic acids dissociate and contribute to the ion balance (Hennigan et al., 2015). Furthermore, they can form salt complexes with inorganic species *e.g.* ammonium oxalate. Neglecting organic acids, as well as other atmospheric species such as $HCO_3^-$ and basic amines, will lead to inaccuracies in the calculated $H^+$.


Thermodynamic equilibrium models such as ISOROPPIA-II can be used as an alternative method for predicting aerosol pH, and whilst these models produce better results when gas phase measurements such as $NH_3$ and $HNO_3$ are available, it is possible to use the measurements obtained during this study to make a prediction of aerosol pH (Fountoukis and Nenes, 2007). This can be achieved using the model in reverse mode, in which known quantities are the concentrations of ammonium, sulfate,

sodium, chloride, nitrate, calcium, potassium and magnesium in the aerosol phase, as well as ambient temperature and relative humidity. The output of the model is the concentration of species in the solid, liquid and gas phase and a prediction of aerosol pH. The ambient temperature and relative humidity data were taken from the measurements made nearby at the Sultan Ismail Petra airport. It is likely that these measurements are representative of the Bachok research station, as further investigation of data from two other meteorological stations revealed that temperature and relative humidity remain consistent along the

coastline. A map to show the location of the three meteorological stations along the east coast, as well as the Bachok research station, can be found in the Supplement (Fig. S4). The two other stations are Narathiwat airport (6.520° N, 101.743° E) and Sultan Mahmud airport (5.383° N, 103.103°E). Average relative humidity between 18-01-2014 and 07-02-2014 was 77.7% at Sultan Petra Ismail airport, 75.4% at Narathiwat airport and 77.0% at Sultan Mahmud airport. Average temperatures recorded at the stations during this time were 24.8, 25.6 and 25.4 °C for Sultan Petra Ismail, Sultan Mahmud and Narathiwat airport

respectively.

The ISOROPPIA-II thermodynamic model predictions of $PM_{2.5}$ pH are shown in Fig. 9. Particle pH was estimated with ISOROPPIA-II run in the reverse mode without gas phase species input, and ranged from -0.97 to 1.12 during the measurement period, implying that the aerosol was highly acidic. The pH prediction for the aerosol collected between midday on 03-02-2014

and 04-02-2014 was 7.06 and has been excluded from Fig. 9. The particle concentrations input on this day correspond to negative values of strong acidity and therefore the model balances charge by assuming $[OH^-] > [H^+]$; this leads to a calculated pH of greater than 7. Murphy et al. have reported that pH prediction is sensitive to strong acidity in the limit of strong acidity approaching zero and that the model can be drastically improved if gas phase $NH_3$ and $HNO_3$ measurements are included. The gas-to-particle partitioning of these species is sensitive to pH under conditions commonly encountered in the atmosphere,

therefore gaseous $NH_3$ and $HNO_3$ measurements provide better constraint on the thermodynamic model.

**Figure 9: Predicted $PM_{2.5}$ pH at the Bachok measurement site using ISOROPPIA-II. Yellow shaded areas represent the time between sunrise and sunset (local).**





Acidic particles can have detrimental effects on human health, air quality, and the health of aquatic and terrestrial ecosystems (Hennigan et al., 2015). For example, Gwynn et al. performed a time-series analysis of acidic PM and daily mortality and morbidity in the Buffalo, New York region; several significant pollutant-health effect associations were identified, the strongest being the correlation between atmospheric $SO_4^{2-}$ concentration and respiratory hospital admissions (Gwynn et al., 2000). Deposition of acidic gases and particles can affect the acid-neutralizing capacity of freshwater ecosystems, leading to

biological damage and loss of invertebrates in worst-case scenarios (Schindler, 1988). The extent to which sulfate aerosol is neutralised also has important implications for aerosol radiative forcing and ice cloud nucleation (Charlson et al., 1987). For example, estimates of the radiative forcing for anthropogenic sulfate aerosol range from -0.26 to -0.82 W m$^{-2}$ (Graf et al., 1997). Particle acidity can influence various atmospheric chemical processes, including $SO_2$ oxidation, halogen chemistry, and the partitioning of ammonia, nitric acid, organic acids and isomeric epoxy diols from isoprene photooxidation (IEPOX)

(Hennigan et al., 2015; Surratt et al., 2010; Lin et al., 2012). In summary, whilst some of the risks associated with aerosol acidity in Bachok originate from local sources of pollution, it is possible that people living in these rural areas are also exposed to an additional risk, as the region appears to be sensitive to the effects of industrialisation further afield in East Asia.

### 3.3.4 Sources and formation of oxalate, $C_2O_4^{2-}$

Oxalic acid is the most abundant dicarboxylic acid in tropospheric aerosol (Sareen et al., 2016). This major water-soluble

organic component can alter the hygroscopicity of aerosols, and can either act as cloud condensation nuclei (CCN), or reduce the surface tension of particles to form CCN (Saxena and Hildemann, 1996; Novakov and Penner, 1993; Facchini et al., 1999; Kerminen, 2001). In this study, oxalate made up 2.6% of the total measured water-soluble ion content. The average concentration was 0.42 µg m$^{-3}$, and throughout the measurement period the concentration ranged from 0.15 to 0.65 µg m$^{-3}$. Interestingly, such levels of oxalate in atmospheric aerosol are typical of urban environments, despite the fact that the Bachok

research station is located in a rural coastal region. For example, Freitas et al. report average oxalate concentrations in TSP at an urban site and a rural site in Londrina City, Brazil of 0.57 and 0.03 µg m$^{-3}$ respectively (Freitas et al., 2012). Other reported oxalate concentrations in urban TSP include measurements of 0.10 – 0.48 µg m$^{-3}$ in Shanghai (Jiang et al., 2011) and 0.27 µg m$^{-3}$ in Tokyo (Sempere and Kawamura, 1994).

To investigate possible oxalate sources and formation pathways, it is necessary to consider the correlation of oxalate with different atmospheric species. Jiang et al. report using $NO_2^-$ as an indicator for vehicle emissions, $nss$$SO_4^{2-}$ and $NO_3^-$ for secondary formation through different pathways, and $K^+$ for biomass burning (Jiang et al., 2011). In this study, there was a relatively strong correlation between oxalate and $nss$$SO_4^{2-}$ (R = 0.60, p < 0.001), suggesting a common formation pathway of the two species. It is well known that $SO_4^{2-}$ forms $via$ aqueous oxidation (Seinfeld and Pandis, 2006), and modelling studies

also suggest that aqueous chemistry is a large contributor of oxalate formation globally (Myriokefalitakis et al., 2011). Furthermore, Carlton et al. report that whilst there are likely to be many sources of oxalate, oxidation of pyruvate in the aqueous



phase is known to form oxalate at dilute (cloud-relevant) concentrations (Carlton et al., 2006). Tan et al. also state that aqueous acetate oxidation is a key source of oxalate (Tan et al., 2012). A positive correlation between oxalate and $NH_4^+$ was also observed (R = 0.66, p < 0.001). A similar observation was reported by Jiang et al. in a study of aerosol oxalate in Shanghai

(Jiang et al., 2011). Using size distribution data, they were able to propose that the correlation was due to the presence of ammonium oxalate in the aerosol. In this study there is no size distribution data available, and so it is important to consider the fact that the correlation may be linked to the influence of sulfate on both $NH_4^+$ and oxalate in aerosol; $NH_4^+$ partitions to the aerosol from gaseous $NH_3$ in an attempt to neutralise acidic sulfate particles, whilst oxalate exhibits similar formation pathways to $SO_4^{2-}$.


A study carried out by Huang et al. in the urban area of Shenzen (Southern China) reported that whilst good correlation of droplet oxalate with $K^+$ was observed ($R^2 = 0.75$, average diameter = 1.0 µm), there was poor correlation between oxalate and $K^+$ in the condensation mode ($R^2 = 0.10$, average diameter = 0.4 µm) (Huang et al., 2006). This implied that whilst biomass burning was probably not an important primary source of condensation mode oxalate, it is likely that biomass burning particles

act as effective CCN, promoting in-cloud sulfate and oxalate formation. In this study, oxalate correlated strongly with $nssK^+$ (R = 0.65, p < 0.001). The oxalate/$nss$K$^+$ ratio was used to predict whether biomass burning was an important primary or secondary source of oxalate. The oxalate/$nss$K$^+$ ratio ranged from 0.49 to 0.99 during the Bachok demonstration campaign, significantly higher than those found in aerosol directly emitted from vegetation fires in the Amazon Basin (Yamasoe et al., 2000). This suggests that biomass burning is an important secondary source of oxalate in the Bachok region, rather than a

significant primary source. A similar hypothesis was proposed by Huang et al., who measured ambient $PM_{2.5}$ in an urban environment in the Pearl River Delta Region of China, and reported oxalate/$K^+$ ratios of 0.57 and 0.33 in summer and winter respectively (Huang and Yu, 2007). However, it is worth noting that whilst the higher ratios reported in both studies indicate that biomass burning is not a major primary source of oxalate, measurements of oxalate and $K^+$ in local biomass burning aerosols (rather than aerosol in the Amazon Basin) would provide a better indication of the source contribution by biomass

burning.

There was no significant correlation observed between oxalate and $NO_3^-$, or between oxalate and $NO_2^-$. The lack of correlation with $NO_3^-$ suggests that the two species do not have similar formation pathways, and that vehicle emissions are not an important secondary source of oxalate. It is also unlikely that vehicle emissions contribute to the primary sources of oxalate, due to the

lack of correlation between oxalate and $NO_2^-$. However, $NO_2^-$ was only detected in 7 out of the 30 samples collected, so it is difficult to ascertain whether there is a relationship between these two species or not.

### 3.3.5 Sea salt aerosol and factors affecting chloride depletion

On average, $Na^+$ and $Cl^-$ contributed 7.0% and 4.1% to the total measured water-soluble ion content respectively, and 72% of the measured $Na^+$ was attributed to $ss$Na$^+$. The concentration of $ss$Na$^+$ ranged from 0.24 to 2.35 µg m$^{-3}$, whilst the concentration



of Cl⁻ ranged from 0.003 to 2.38 µg m⁻³. There was no correlation between $nss$Na⁺ and Cl⁻ (R = 0.01), but a strong positive correlation between $ss$Na⁺ and Cl⁻ was observed (R = 0.83, p < 0.001). During the measurement period, the Cl⁻/$ss$Na⁺ molar ratio ranged from 0.003 to 1.10 with an average value of 0.40. A time series of Cl⁻/$ss$Na⁺ molar ratio can be found in the Supplement (Fig. S5). All of the ratios recorded were lower than that of bulk seawater, 1.18 (Boreddy and Kawamura, 2015). Figure 10 shows backward air mass trajectories arriving at the Bachok research station, coloured by the Cl⁻/$ss$Na⁺ molar ratio.

The plot shows that the lowest Cl⁻/$ss$Na⁺ ratios are found when continental air masses from highly industrialised countries such as China and Vietnam arrive at the site, and higher Cl⁻/$ss$Na⁺ ratios are found when marine air masses from the South China Sea arrive.

**Figure 10: 10-day HYSPLIT back trajectories centred on the Bachok research station, between 18-01-2014 and 07-02-2014. The**
**back trajectories are coloured by the Cl⁻/$ss$Na⁺ molar ratio. Plot constructed using the openair package in RStudio (Carslaw and Ropkins, 2012; Carslaw, 2015).**

The relationship between Cl⁻/$ss$Na⁺ molar ratio and air mass origin shown in Fig. 10 clearly indicates that the extent of chloride depletion is greater when the aerosol has been more influenced by anthropogenic sources of pollution. A study of marine
aerosol at remote Chichijima Island in the western North Pacific during 2001 and 2002 reported that mean Cl⁻/Na⁺ molar ratios were highest (1.34) in September 2001 and lowest (0.30) in May 2002, and the mean ratio across the 2-year period was 1.10 (Boreddy et al., 2014). Boreddy et al. reported that the observed chloride depletion was likely due to acid displacement occurring as a result of atmospheric mixing of anthropogenic pollutants such as SO$_x$ and NO$_x$. Acid displacement can occur when sea salt particles react with acids such as $H_2SO_4$, $HNO_3$, oxalic acid ($C_2H_2O_4$) and methanesulfonic acid ($CH_3SO_3H$) in
the atmosphere. Such processes are of atmospheric importance as they lead to the formation of gaseous HCl and potentially affect acid deposition conditions in the region. In a more recent study, Boreddy and Kawamura performed regression analysis between the Cl⁻/Na⁺ mass ratio and various acidic species, including $nss$SO$_4^{2-}$, NO$_3^-$, MSA⁻ and oxalic acid. They found a moderate negative correlation between Cl⁻/Na⁺ mass ratio and $nss$SO$_4^{2-}$, and a moderate to weak negative correlation with NO$_3^-$; this suggested that sulfate had a higher influence on chloride depletion than nitrate. Furthermore, whilst MSA⁻
moderately correlated with the Cl⁻/Na⁺ mass ratio in the summer, there was significant negative correlation between oxalic acid and the mass ratio in the other three seasons, providing confirmation that oxalic acid plays an important role in chloride loss at Chichijima Island. A similar regression analysis was carried out on the measurements obtained during the Bachok measurement campaign, and whilst a weak negative correlation was found between SO$_4^{2-}$ and the Cl⁻/$ss$Na⁺ mass ratio (R = 0.41, p < 0.001), there was no correlation of the Cl⁻/$ss$Na⁺ mass ratio with oxalate or MSA⁻. These results imply that
whilst $H_2SO_4$ played an important role in chloride depletion, methanesulfonic acid and oxalic acid may not have done. Interestingly, a strong positive correlation was observed between NO$_3^-$ and Cl⁻/$ss$Na⁺ mass ratio (R = 0.78, p < 0.001). This correlation may be explained by the fact that the high sulfate content of the aerosol in Bachok is capable of suppressing both NO$_3^-$ and Cl⁻ levels.



### 3.3.6 Using $nss$Ca$^{2+}$ as a potential tracer for dust episodes

$Nss$Ca$^{2+}$ can be used as a tracer for atmospheric dust (Boreddy and Kawamura, 2015). The average concentration of $nss$Ca$^{2+}$ during the measurement campaign was 0.07 µg m$^{-3}$ and the maximum concentration recorded was 0.28 µg m$^{-3}$. On most days, the $nss$Ca$^{2+}$ concentration was below 0.10 µg m$^{-3}$, but between midday on 30-01-2014 and midnight on 31-01-2014 (local times), elevated $nss$Ca$^{2+}$ levels were observed and the average concentration during this period was 0.22 µg m$^{-3}$. The same trend was observed for PO$_4$$^{3-}$; the average concentration across the entire measurement period was 0.34 µg m$^{-3}$, but during this 530 12-hour episode the PO$_4$$^{3-}$ concentration increased to 1.88 µg m$^{-3}$. The increase in ion concentration was less pronounced for $nss$Na$^+$ and Mg$^{2+}$, but the concentrations were still 0.26 and 0.88 µg m$^{-3}$ above the average for the whole measurement period respectively. Time series plots for $nss$Ca$^{2+}$, PO$_4$$^{3-}$, Mg$^{2+}$ and $nss$Na$^+$ can be found in the Supplement (Figure S6).

Between midday on 30-01-2014 and midnight on 31-01-2014, $nss$Ca$^{2+}$ contributed significantly (1.26%) to the total measured 535 water-soluble ion content compared to the average contribution during the remainder of the measurement period (0.35%). The Bachok research station is located in the outflow region of Asian dusts, and these measurements suggest that long-range transport of Asian dusts over the measurement site has occurred during this time. In fact, during this episode, the back trajectories arriving at the site can be traced back to the North China Plains and the Horqin Sandy Land in East China; source attribution studies by Ginoux et al. have revealed that large anthropogenic dust sources are found in these regions (Ginoux et 540 al., 2012). In the future, it is important that longer term measurements are carried out at the Bachok research station, to provide confirmation that dust episodes in these regions of Asia are responsible for the elevated levels of $nss$Ca$^{2+}$ and other associated ions. Dust is one of the most abundant types of aerosol in the atmosphere and can have important impacts on both air quality and climate, therefore it is important that seasonal and annual trends in water-soluble ions are studied in more detail at the Bachok research station.

### 545 4 Conclusions

An accurate and reliable technique relying on ion chromatography has been used to make time-resolved measurements of water-soluble ions in atmospheric aerosol at the Bachok Marine and Atmospheric research station. Using meteorological data from the nearby airport, and HYSPLIT backward air mass trajectories centred on the Bachok research station, it was possible to observe both the diurnal wind pattern behaviour, and assess where the air masses arriving at the site originated from. Air 550 quality at this remote location is influenced by local anthropogenic and biogenic emissions, as well as marine air masses from the South China Sea and aged emissions transported from highly polluted East Asian regions during the winter monsoon season. In general, the site was influenced by south westerlies coming from the land from the early hours of the morning until approximately 11 am, and then a dramatic shift in wind direction occurred and a sea breeze was present for the remainder of the day. This shift was accompanied by a drop in the concentrations of NO$_x$ and anthropogenic VOCs (Dunmore et al., 2016).

555



Twelve atmospheric water-soluble ions were measured in this study and $SO_4^{2-}$ was found to be the most dominant ion present, making up 66% of the total measured ion content on average. The non-sea salt and sea salt components of $SO_4^{2-}$, $Na^+$, $K^+$ and $Ca^{2+}$ were determined, and it was found that 96% of the measured $SO_4^{2-}$ was non-sea salt $SO_4^{2-}$. Predictions of aerosol pH were made using the ISOROPPIA-II thermodynamic model and it was estimated that the aerosol was highly acidic, with pH values ranging from -0.97 to 1.12; such levels of acidity are likely to have a detrimental impact on human health and the health of ecosystems at this remote coastal location. A clear difference in aerosol composition was found between continental air masses originating from industrialised regions of East Asia and marine air masses predominantly influenced by the South China Sea. For example, elevated $SO_4^{2-}$ concentrations were observed when continental air masses that had passed over highly industrialised regions of East Asia arrived at the measurement site.

Correlation analyses amongst ionic species and assessment of ratios between different ions provided an insight into common sources and formation pathways of key atmospheric ions. Oxalate concentrations were recorded and found to be more comparable to measurements made at urban locations rather than rural ones. A strong correlation of $C_2O_4^{2-}$ with $SO_4^{2-}$ suggested a common aqueous oxidation formation pathway. Strong correlation between $C_2O_4^{2-}$ and $K^+$ coupled with high $C_2O_4^{2-}/nssK^+$ ratios indicated that biomass burning was an important secondary source of oxalate in the Bachok region, whereas lack of correlation with $NO_2^-$ and $NO_3^-$ suggested that vehicular emissions were not an important source. The average $Cl^-/ssNa^+$ molar ratio during the measurement campaign was 0.40, significantly lower than that of bulk seawater (1.18). Analysis of back trajectories revealed that chloride depletion was greater when the aerosol was more influenced by anthropogenic sources of pollution. Elevated levels of $nssCa^{2+}$ and other ions such as $PO_4^{3-}$, $Mg^{2+}$ and $nssNa^+$ were observed between midday on 30-01-2014 and midnight on 31-01-2014. Assuming that $nssCa^{2+}$ can be used as a tracer for atmospheric dust, it was proposed that the increased concentrations were a result of air masses arriving at the site from the North China Plains and Horqin Sandy Lands. Longer term measurements are required to fully investigate the influence of Asian dusts at this remote coastal location.

To our knowledge, time-resolved measurements of water-soluble ions in $PM_{2.5}$ are virtually non-existent in rural locations on the east coast of Peninsular Malaysia. The data presented in this study has demonstrated the capabilities of the new atmospheric tower at the Bachok research station and has provided an initial insight into factors affecting aerosol composition on this coastline. In the future, it is important that longer term measurements are carried out, with increased time-resolved sampling and particle size fractionation, to provide a better understanding of the factors affecting aerosol composition at this measurement site. This remote location is susceptible to the effects of local, regional and international air pollution, and rapid industrialisation in East Asia is influencing air quality along the east coast of Peninsular Malaysia.

*Data availability.* Raw data is available on PURE (DOI: 10.15124/bd4a9045-832b-4ff8-aecc-ef1653603f1d).



*Author contribution.* All authors contributed to the final version of this article. Naomi J Farren analysed the aerosol samples
and wrote the paper under the supervision of Jacqueline F Hamilton. Rachel E Dunmore collected the aerosol samples. M Iqbal
Mead, Mohd Shahrul Mohd Nadzir, Azizan Abu Samah and Siew-Moi Phang coordinate and manage the University of Malaya
BMRS. William T Sturges coordinated the Bachok demonstration 'International Opportunities Fund' campaign.

*Competing interests.* The authors declare that they have no conflict of interest.

*Acknowledgements.* The financial support of the Natural Environment Research Council (N. Farren, PhD studentship
NE/L501751/1) is gratefully acknowledged. N. Farren would like to thank David Carslaw for his assistance using Openair.
All authors would like to acknowledge NERC (NE/J016012/1 and NE/J016047/1) for funding the Bachok demonstration
'International Opportunities Fund' campaign and HICoE-MoHE IOES-2014 (Air-Ocean-Land Interactions) for supporting the
Bachok Marine Research Station facilities.

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

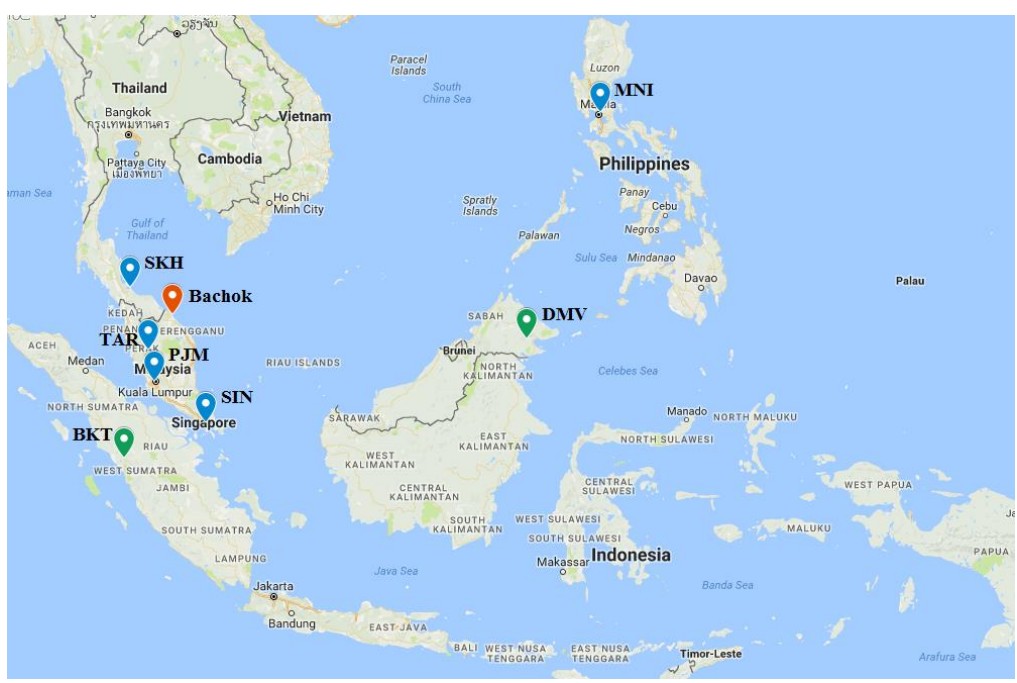

**Figure 1: Location of global (green pins) and regional (blue pins) GAW sites in the Maritime Continent; Danum Valley in Malaysia**
**(DMV), Bukit Kototabang in Indonesia (BKT), Manila in Philippines (MNI), Songkhla in Thailand (SKH), Tanah Rata in Malaysia (TAR), Petaling Jaya in Malaysia (PJM) and Singapore (SIN) (gawsis.ch, 2017). The red pin shows the location of the Bachok Marine and Atmospheric Research Station. Map created using google maps (google.com, 2017).**




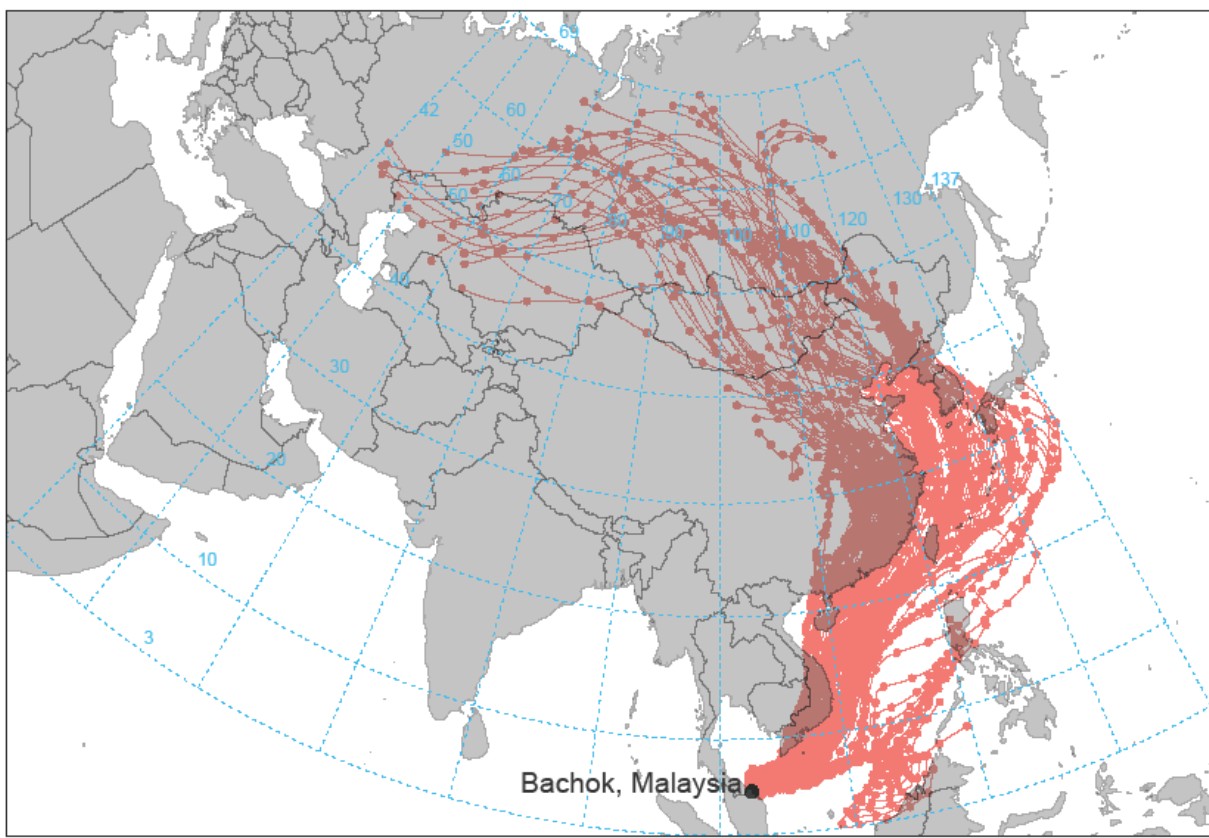

**Figure 2: 10-day HYSPLIT backward air mass trajectories centred on the Bachok Marine and Atmospheric Research Station between 18-01-2014 and 07-02-2014. Plot constructed using the openair package in RStudio (Carslaw and Ropkins, 2012; Carslaw, 2015).**



**Table 1: Mean and maximum ion concentrations measured throughout the measurement period. The average % mass contribution of each ion to the total measured ions is included, as well as the % of samples in which each target ion is found (%Qt).**

| Ion | Mean [ion] / µg m⁻³ | Maximum [ion] / µg m⁻³ | Mean % mass of total measured ion content | %Qt[a] | %RSD$_{total}$[b] |
|---|---|---|---|---|---|
| $SO_4^{2-}$ | 10.7 | 20.8 | 65.6 | 100 | 11.2 |
| $NH_4^+$ | 1.69 | 4.73 | 10.4 | 100 | 6.38 |
| $Na^+$ | 1.13 | 2.60 | 6.95 | 100 | 6.88 |
| $Cl^-$ | 0.67 | 2.38 | 4.14 | 100 | 8.49 |
| $NO_3^-$ | 0.61 | 1.52 | 3.76 | 100 | 22.6 |
| $C_2O_4^{2-}$ | 0.42 | 0.65 | 2.57 | 97 | 13.9 |
| $PO_4^{3-}$ | 0.36 | 2.34 | 2.22 | 93 | 15.4 |
| $K^+$ | 0.38 | 0.67 | 0.67 | 100 | 6.35 |
| $Ca^{2+}$ | 0.10 | 0.35 | 0.64 | 100 | 9.26 |
| $Mg^{2+}$ | 0.10 | 0.21 | 0.61 | 100 | 6.72 |
| $CH_3SO_3^-$ | 0.08 | 0.22 | 0.47 | 67 | 10.6 |
| $NO_2^-$ | 0.05 | 0.16 | 0.33 | 23 | 14.3 |
| **Total** | **16.2** | **27.0** | -- | -- | -- |

[a]Percentage of samples in which the target ion was above the LOQ. [b]Total error associated with each ion.








**Figure 3: Time series of SO₄²⁻, NH₄⁺, Na⁺, Cl⁻, NO₃⁻ and NO₂⁻ concentration (µg m⁻³) during the Bachok demonstration campaign (18-01-2014 to 07-02-2014). Yellow shaded areas represent the time between sunrise and sunset (local).**




**Figure 4: Time series of PO₄³⁻, Ca²⁺, Mg²⁺, K⁺, C₂O₄²⁻ and CH₃SO₃⁻ concentration (µg m⁻³) during the Bachok demonstration campaign (18-01-2014 to 07-02-2014). Yellow shaded areas represent the time between sunrise and sunset (local).**





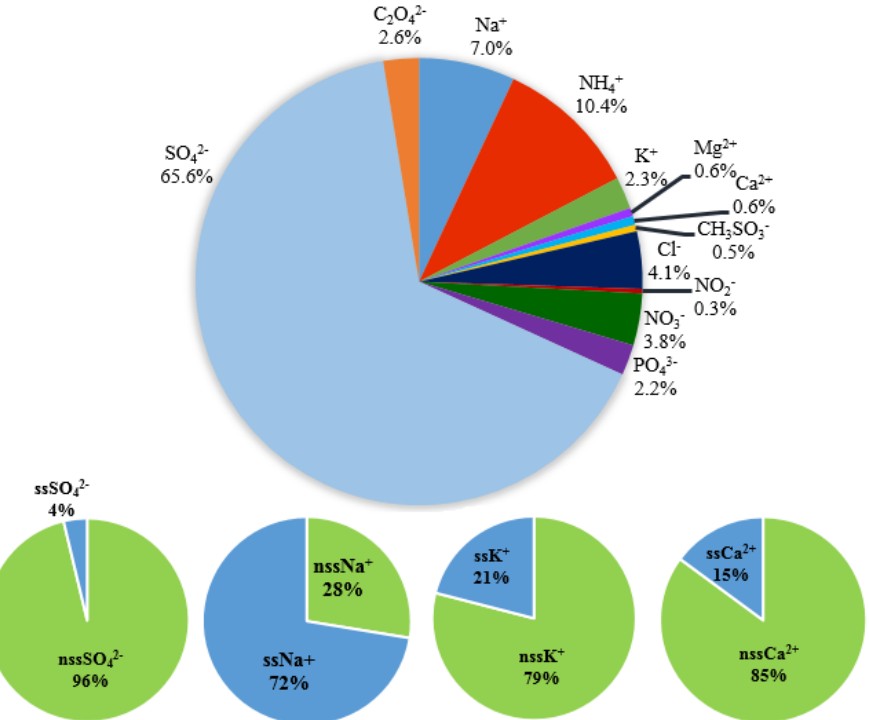

**Figure 5: Pie chart to show the average mass composition of water-soluble ions in aerosol collected at the Bachok research station (upper panel) and pie charts to show the percentage of non-sea salt and sea salt fractions of Na$^+$, SO$_4^{2-}$, K$^+$, Ca$^{2+}$ (lower panel, left to right).**





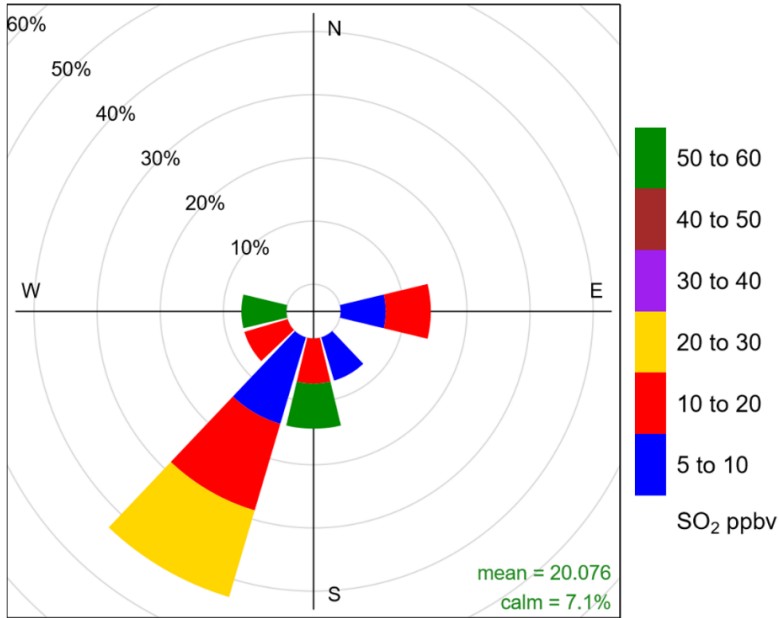

**Frequency of counts by wind direction (%)**

**Figure 6: Pollution rose to show the relationship between wind direction and SO₂ concentration (≥ 5 ppb) at the Bachok measurement site. Plot constructed using the openair package in RStudio (Carslaw and Ropkins, 2012; Carslaw, 2015).**








**Figure 7: Upper panel shows 10-day HYSPLIT backward air mass trajectories centred on the Bachok research station between 18-01-2014 and 07-02-2014. The back trajectories are coloured by the concentration of SO₄²⁻ (µg m⁻³). Lower panel shows the 10-cluster solution to backward air mass trajectories centred on the Bachok research station during the same time period. The clusters are coloured by the average concentration of SO₄²⁻ (µg m⁻³). Plot constructed using the openair package in RStudio (Carslaw and Ropkins, 2012; Carslaw, 2015).**




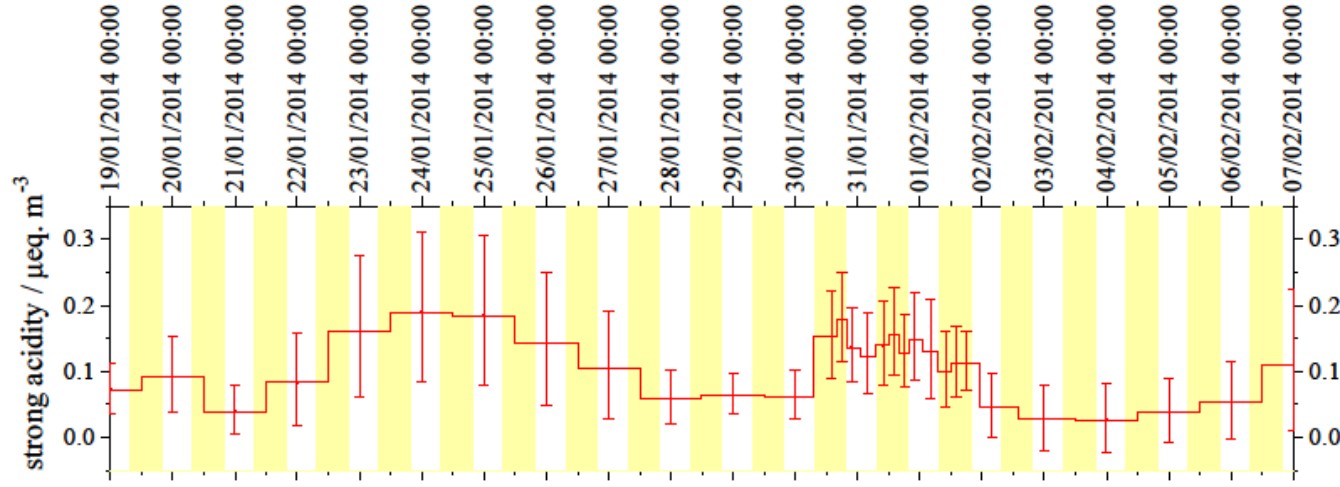

**Figure 8: Particle strong acidity and associated error predictions for the aerosol collected during the Bachok measurement campaign. Yellow shaded areas represent the time between sunrise and sunset (local).**

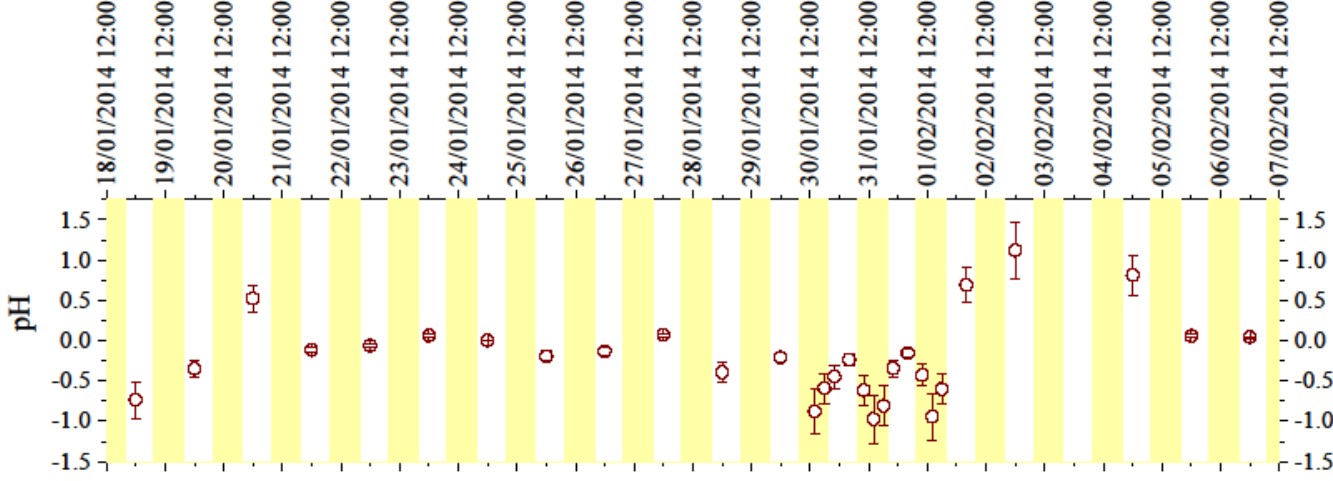


**Figure 9: Predicted PM$_{2.5}$ pH at the Bachok measurement site using ISOROPPIA-II. Yellow shaded areas represent the time between sunrise and sunset (local).**






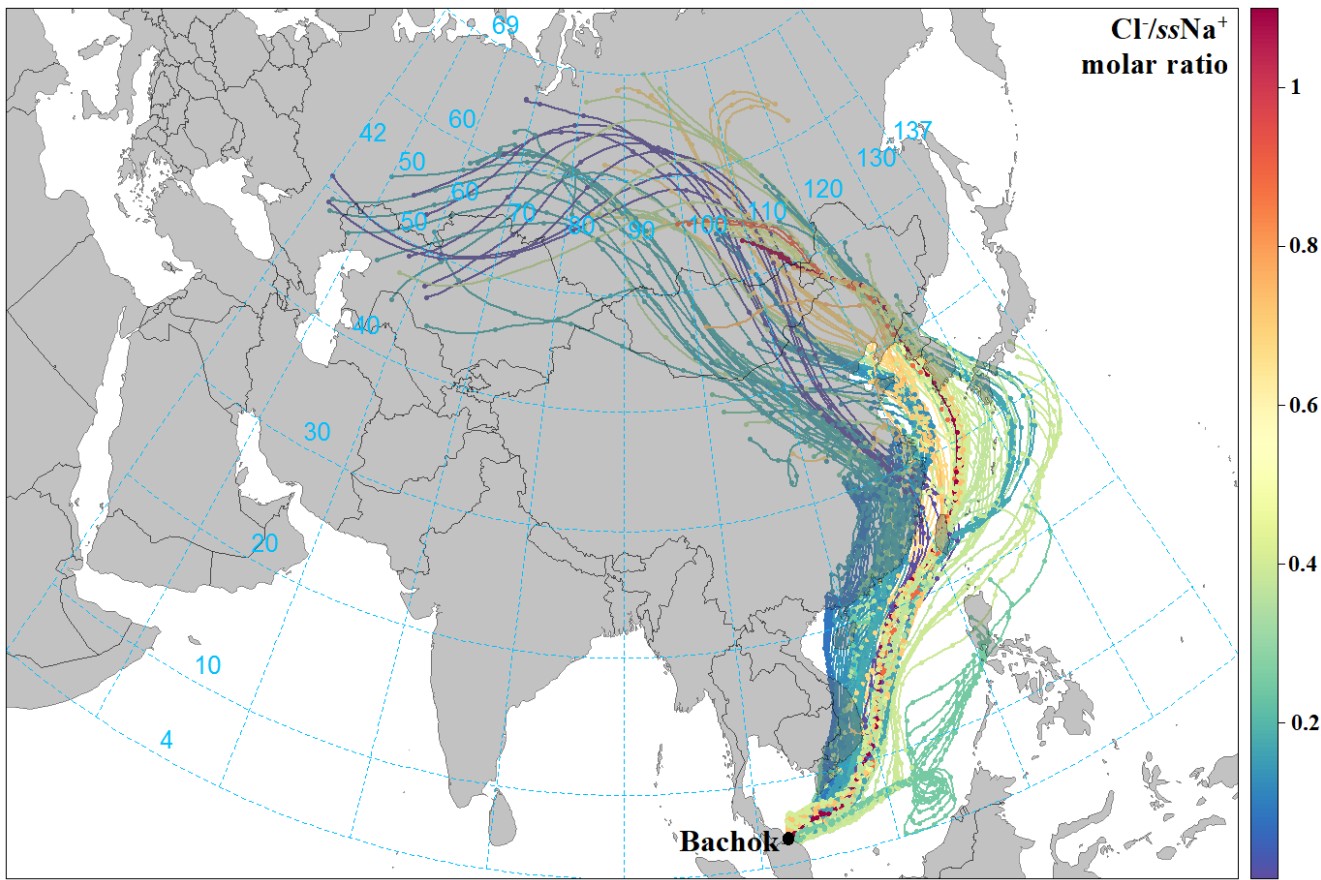


**Figure 10: 10-day HYSPLIT back trajectories centred on the Bachok research station, between 18-01-2014 and 07-02-2014. The back trajectories are coloured by the Cl⁻/ssNa⁺ molar ratio. Plot constructed using the openair package in RStudio (Carslaw and Ropkins, 2012; Carslaw, 2015).**