# Peer review of "Chemical Characterisation of Water-soluble Ions in Atmospheric Particulate Matter on the East Coast of Peninsular Malaysia"

_Atmospheric Chemistry and Physics, 2018_

## Referee Comment (RC1) · Anonymous Referee #2 · 13 Jun 2018

The manuscript "Chemical Characterisation of Water-soluble Ions in Atmospheric Particulate Matter on the East Coast of Peninsular Malaysia" by Farren et al. investigated the particulate matters on the east coast of Peninsular Malaysia. Chemical components of particles (mainly soluble ions) were measured. Air mass trajectories were applied to indicate the potential source regions of various aerosol components. A thermodynamic model is used to estimate the aerosol acidity. Generally, this study revealed the characteristics of atmospheric chemistry over the East Coast of Peninsular Malaysia, which has been rarely reported. This manuscript served to fill in the gap of the Southeast Asia region which has been poorly characterized of its emissions, air quality, etc.

[Figure]

However, the chemical characteristics of aerosol over this region is not well studied. The design of the measurement is inadequate based on a Jan. – Feb. sampling of about thirty samples. No sampling during the biomass burning season is conducted. The presentation of the data analysis (almost all the figures) is not good. Substantial revisions are suggested before this manuscript can be further reviewed.

The major comments are list below: 1. Section 2: The method section should be re-organized. The sampling part should be described in the beginning of the section. Then the analytical procedures are presented. This study used quartz fiber filters for particle collection and the subsequent ion analysis. However, it is known that the quartz fiber filters have high background values of some cations such as $Ca2+$, $Mg2+$, etc. Did the authors perform ion analysis of the blank filters? What are the values of the ions of the blank filters?

Section 3.1 should be moved to the methodology section as it is related to the uncertainties of the chemical analysis but not the analysis results.

Line 395 – 405: The description of ISOROPPIA-II should be moved to the Method Section.

2. Line 222 – 224: It is not appropriate to compare the results with the previous one as the study period is quite different. Furthermore, why don't use the concentrations of PM2.5 based on your data? 3. Line 269 – 270: Is there any volcano activity during the study period. If not, this citation is not necessary. 4. Line 287 – 288: The pollution rose plot (Fig. 6) cannot show the SO2 concentrations under calm conditions as the rose plot is based on conditions with wind speed higher than zero. Thus, the writings "The majority of higher SO2 events were observed in calm conditions when the air arriving at the site had passed over land to the south west of Bachok." is not based on sound analysis. 5. Line 310 – 320: It is concluded that clusters 4 and 10 are associated with high SO42- concentrations of 20.4 and 18.1 $\mu$g m-3, respectively. It is understandable that cluster 4 passed over industrialized areas such as southern

China and Southeast Asia, thus bringing considerable amounts of sulfate. However, it is explained that the high sulfate in cluster 10 is attributed to Manila First, it should be noted that the total emissions and emission intensity of Manila should be much lower than mainland China. Secondly, Cluster 10 travelled long distances over the ocean, which is supposed to have a clean effect on the aerosol concentrations.

6. What is the definition of chlorine-containing very short-lived substances (Cl-VSLSs)? What species are included as Cl-VSLSs? It seems that the authors regarded Cl-VSLSs as a tracer for anthropogenic emissions and use it for further analysis of $SO_4^{2-}$ during the pollution and less polluted periods. This is problematic as $SO_4^{2-}$ and Cl-VSLSs should have different origins and behavior during the long-range transport, e.g. dry/wet deposition, decomposition rate.

7. Line 430 – 437: These paragraphs are basically not related to this study.

8. Line 474 – 475: As indicated by Fig. 4, the levels of $K^+$ were less than 1ug/m3 throughout the whole study period, suggesting no significant biomass burning events. Thus, it is wrongly concluded that "biomass burning is a secondary source of oxalate".

9. Section 3.3.5: This section discussed about the $Cl^-/Na^+$ ratio and found the ratio was lower than its value of the seawater. It is concluded that the anthropogenic pollution via the long-range transport (Fig. 10) exerted the impact on the depletion of chloride. This is questionable as the study period is Jan. – Feb., which is the winter heating season in China. Based on Fig. 10, the air masses passed over vast areas of northern China, indicating aerosol rich in chloride from coal burning should be derived. If the monitoring site is influenced by emissions from China as discussed by this study, the $Cl^-/Na^+$ ratio should be much higher than 1.0.

10. Line 522 – 523: It is hard to say that sulfate suppressed the formation of nitrate. The possible cause should be the deficiency of NH3, leading to the incomplete neutralization of sulfate and nitrate.

[Figure]

Minor Comments:

Page 6, Line 191: Taiwan is not a country.

Line 204: Ashfold et al. ; Line 222: Dominick et al.; Line 322: Oram et al.; Line 445: Freitas et al.; Line 456: Carlton et al.; Line 466: Huang et al.; The format is incorrect. Line 335: what does "a NAME trajectory" mean?

---

## Referee Comment (RC2) · Anonymous Referee #1 · 15 Jun 2018

The manuscript by Farren et al. entitled as 'Chemical Characterisation of Water-soluble Ions in Atmospheric Particulate Matter on the East Coast of Peninsular Malaysia' presents the observation data at Malaysia. The method and data quality seem to be reasonably good. The data in the manuscript could be a good addition to the existing data set in the region. Quality of figures and descriptions could significantly be improved. I provide some comments related to the presentation quality below. It would be good if the authors could significantly improve it.

Comments

L45 'During the northern hemisphere winter, a large anticyclone forms over Siberia

each year, creating strong north-easterly monsoon winds in the South China Sea (Northeast Monsoon). These strong north-easterlies can transport air masses from rapidly developing East Asian countries (e.g. China, Japan, Taiwan, Vietnam, North and South Korea) across the South China Sea to the Maritime Continent.'

I am not sure if the statement is true. Please add references to support the description.

L185 'Figure 2 shows the 10-day backward air mass trajectories arriving at the measurement site during the demonstration campaign.'

Further details of the back-trajectory calculations, such as altitude, will be needed. It is not clear to me if a back trajectory analysis in the troposphere could provide a reliable result for such a long time-scale.

L224 'and it is likely that the remainder was comprised primarily of organic aerosol'

Please provide a supporting information on this statement.

L240 'the mean Na+ /Ca2+ ratio in the crust and mean Ca2+ /Na+ ratio in seawater have been estimated as 1.78 w/w and 0.038 w/w respectively (Bowen, 1979)' I wonder how stable these values are. The uncertainties in the values directly influence the following discussion. Please provide a detailed description, rather than simply referring one publication.

L314 'Air masses in cluster 10 passed over the megacity of Manila in the Philippines, but may have slightly lower SO42- levels due to the height of the back trajectories;'

I am unable to judge if the statement is valid, as no information about altitude is provided in the manuscript.

L351 'The uptake of SO42- is preferential to the uptake of NO3- because sulfuric acid has a lower vapour pressure than nitric acid, and aqueous or solid (NH4)2SO4 is the preferred form of sulfate'

The statement is unclear to me. Please clarify.

L402 ‘The ambient temperature and relative humidity data were taken from the measurements made nearby at the Sultan Ismail Petra airport.’

It seems to me that the authors assumed an internal mixing state in using the thermodynamic model. Is there any supporting evidence on this assumption?

Figure 7

Almost all the trajectories look similar to me, except for C10. Pleas provide the detailed reasoning for classfication.

Minor comments

L76 ‘Dominick et al. characterised ..’

I believe that it should be written as ‘Dominick et al. (2015) characterized. . .’ There are many similar descriptions when the authors cite other publications. Please check the recent publications of the journal carefully in preparing a manuscript.

L200 ‘The station is located approximately 23 km away at the Sultan Ismail Petra airport in Kota Bharu (6.17298N, 102.2928E), as shown in Fig. S1 (Supplement)’

A similar information has already appeared at L150. Please minimize duplicated descriptions.

---

## Author Comment (AC1) · 31 Jul 2018

**Response to referees: Chemical Characterisation of Water-soluble Ions in Atmospheric Particulate Matter on the East Coast of Peninsular Malaysia**

The authors would like to thank each referee for their positive remarks about the paper and their interesting suggestions. The specific comments are addressed below.

**Anonymous Referee #2**

The manuscript "Chemical Characterisation of Water-soluble Ions in Atmospheric Particulate Matter on the East Coast of Peninsular Malaysia" by Farren et al. investigated the particulate matters on the east coast of Peninsular Malaysia. Chemical components of particles (mainly soluble ions) were measured. Air mass trajectories were applied to indicate the potential source regions of various aerosol components. A thermodynamic model is used to estimate the aerosol acidity. Generally, this study revealed the characteristics of atmospheric chemistry over the East Coast of Peninsular Malaysia, which has been rarely reported. This manuscript served to fill in the gap of the Southeast Asia region which has been poorly characterized of its emissions, air quality, etc.

However, the chemical characteristics of aerosol over this region is not well studied. The design of the measurement is inadequate based on a Jan. – Feb. sampling of about thirty samples. No sampling during the biomass burning season is conducted.

Whilst we agree that our study is over a relatively short period, we feel that this work gives sufficient insight into the factors that affect aerosol composition to be suitable for publication in ACP. This study was not targeted at biomass burning but instead the impact of pollution outflow from Southeast Asia during very polluted periods.

The presentation of the data analysis (almost all the figures) is not good. Substantial revisions are suggested before this manuscript can be further reviewed. The major comments are list below: 1. Section 2: The method section should be reorganized. The sampling part should be described in the beginning of the section. Then the analytical procedures are presented. This study used quartz fiber filters for particle collection and the subsequent ion analysis. However, it is known that the quartz fiber filters have high background values of some cations such as Ca2+, Mg2+, etc. Did the authors perform ion analysis of the blank filters? What are the values of the ions of the blank filters?

The sample and extraction paragraph of the method section has been moved to the beginning of the section, followed by the analytical procedures.

As mentioned in section 2.4 (method validation), ion analysis of the blank filters has already been performed as part of this study. Blank subtractions were applied to any target ions found in detectable amounts. Procedural blank peak areas for each ion and average blank contribution to field samples over the entire sampling period are provided in the Supplement, Table S1.

Section 3.1 should be moved to the methodology section as it is related to the uncertainties of the chemical analysis but not the analysis results.

Section 3.1, which relates to the uncertainties of the chemical analysis, has been incorporated into section 2.4 (method validation).

Line 395 – 405: The description of ISOROPPIA-II should be moved to the Method Section.

*The description of ISOROPPIA-II has been integrated into the method section (Section 2.6, ISOROPPIA-II model) and section 3.3.3 has been reworded accordingly.*

2. Line 222 – 224: It is not appropriate to compare the results with the previous one as the study period is quite different. Furthermore, why don't use the concentrations of PM2.5 based on your data?

*Concentrations of $PM_{2.5}$ based on our data are not available. These sentences were written to introduce the section on aerosol composition but are not essential for the study and have been removed.*

3. Line 269 – 270: Is there any volcano activity during the study period. If not, this citation is not necessary.

*There was no known volcanic activity during the study period. This sentence has been removed.*

4. Line 287 – 288: The pollution rose plot (Fig. 6) cannot show the SO2 concentrations under calm conditions as the rose plot is based on conditions with wind speed higher than zero. Thus, the writings "The majority of higher SO2 events were observed in calm conditions when the air arriving at the site had passed over land to the south west of Bachok." is not based on sound analysis.

*This observation has now been explained more carefully (lines 347-352). Figure 6 shows the relationship between wind direction and $SO_2$ concentration for $SO_2$ concentrations $\geq$ 5 ppb. In the bottom right corner, 'mean' represents the mean $SO_2$ concentration (20.1 ppb) and 'calm' represents the fraction of data that cannot be attributed to a specific wind direction (7.1%). The average wind speed during elevated ($\geq$ 5 ppb) $SO_2$ periods was 1.1 m s$^{-1}$. The average wind speed during the lower ($<$ 5 ppb) $SO_2$ periods was considerably higher, 2.8 m s$^{-1}$.*

5. Line 310 – 320: It is concluded that clusters 4 and 10 are associated with high SO42- concentrations of 20.4 and 18.1 µg m-3, respectively. It is understandable that cluster 4 passed over industrialized areas such as southern China and Southeast Asia, thus bringing considerable amounts of sulfate. However, it is explained that the high sulfate in cluster 10 is attributed to Manila First, it should be noted that the total emissions and emission intensity of Manila should be much lower than mainland China. Secondly, Cluster 10 travelled long distances over the ocean, which is supposed to have a clean effect on the aerosol concentrations.

*In response to comments from referee #1, the cluster analysis has been reduced to 5 clusters and 7-day trajectories are reported (rather than 10-day) as they are more representative of the troposphere. Further analysis of this data shows that clusters 2, 3 and 5 are associated with the highest $SO_4^{2-}$ concentrations, 14.4, 13.8 and 18.1 µg m$^{-3}$ respectively. Clusters 2 and 3 passed over industrialised regions e.g. southern China and brought considerable amounts of sulfate. However, without additional measurements, the full reason for the high $SO_4^{2-}$ levels in cluster 5 (same as cluster 10 in referee comment) is not clear. Interestingly, air masses within cluster 5 come from much higher altitudes than the other clusters and there is evidence of a low pressure system, generating anticlockwise winds around a possible cyclone in the South China Sea north of the island of Borneo. Cluster 5 only incorporates the final 24-hour period of the measurement campaign (06/02/2014 12:00 – 07/02/2014 12:00) and only one $SO_4^{2-}$ measurement is available (18.1 µg m$^{-3}$). Further measurements of air masses similar to those in cluster 5 are needed to understand the high $SO_4^{2-}$ levels. This section has been amended to provide a better explanation (lines 381-392).*

6. What is the definition of chlorine-containing very short-lived substances (Cl-VSLSs)? What species are included as Cl-VSLSs? It seems that the authors regarded Cl-VSLSs as a tracer for anthropogenic

emissions and use it for further analysis of SO42- during the pollution and less polluted periods. This is problematic as SO42- and Cl-VSLSs should have different origins and behaviour during the long-range transport, e.g. dry/wet deposition, decomposition rate.

Cl-VSLSs are ozone depleting species with short atmospheric lifetimes, typically less than 6 months. Species include dichloromethane ($CH_2Cl_2$) and 1,2-dichloroethane ($CH_2ClCH_2Cl$). The author agrees that $SO_4^{2-}$ and Cl-VSLSs are likely to have different sources and atmospheric behaviour and that a direct comparison of these pollutants is not suitable. Nevertheless, this section is intended to demonstrate that there is additional evidence to support the observation that polluted air masses, containing a range of chemical pollutants, are being transported from East Asia to tropical regions of the western Pacific. This section (lines 406-419) has been rephrased to explain this more clearly.

7. Line 430 – 437: These paragraphs are basically not related to this study.

This paragraph puts the study into a wider context but has been shortened significantly (lines 520-530).

8. Line 474 – 475: As indicated by Fig. 4, the levels of K+ were less than 1ug/m3 throughout the whole study period, suggesting no significant biomass burning events. Thus, it is wrongly concluded that "biomass burning is a secondary source of oxalate".

Whilst the $K^+$ levels suggest that the measurement site was not heavily influenced by significant biomass burning events in the local area, the levels of $K^+$ do not rule out the influence of biomass burning on a regional/national scale. Biomass burning aerosol in the wider region will have undergone atmospheric processing and dispersion prior to arrival at the measurement site, potentially lowering $K^+$ levels and raising the concentration of secondary species. Due to both the strong correlation of oxalate and $nssK^+$, and the high oxalate/$nssK^+$ ratio, it is likely that biomass burning in the wider region has influenced the secondary formation of oxalate for example. The strong correlation with $SO_4^{2-}$ and weak correlation with $NO_3^-$ suggests that secondary oxalate formation has occurred through an aqueous phase oxidation process, for which biomass burning particles may have acted as cloud condensation nuclei. This section has been reworded to explain this point more clearly (lines 547-562).

9. Section 3.3.5: This section discussed about the Cl-/Na+ ratio and found the ratio was lower than its value of the seawater. It is concluded that the anthropogenic pollution via the long-range transport (Fig. 10) exerted the impact on the depletion of chloride. This is questionable as the study period is Jan. – Feb., which is the winter heating season in China. Based on Fig. 10, the air masses passed over vast areas of northern China, indicating aerosol rich in chloride from coal burning should be derived. If the monitoring site is influenced by emissions from China as discussed by this study, the Cl-/Na+ ratio should be much higher than 1.0.

It is possible that aerosol, particularly in northern China, may be rich in chloride in the winter due to coal burning. For example, our recent measurements at a ground-level urban background site in Beijing during Nov-Dec 2016 show average $Cl^-$ levels of 6.3 µg m$^{-3}$. This is significantly higher than $Cl^-$ concentrations at the Bachok measurement site, which were 0.67 µg m$^{-3}$ on average during the measurement period. However, the air masses passed over regions of northern China at relatively high altitudes, exceeding 4000 m in some cases. In fact, figure 11 has been updated to show 7-day trajectories instead of 10-day trajectories in response to other referee comments, as these are more representative of the troposphere.

As shown in Figure 11, $Cl^-/ssNa^+$ molar ratios were less than 1.18 in all cases, showing significant $Cl^-$ depletion. Greater $Cl^-$ depletion was observed in continental air masses arriving at the site that had passed over industrialised regions in southern China, and Vietnam. The $Cl^-$ depletion process has been

studied extensively and it is widely accepted that Cl⁻ depletion can occur through the volatilization of HCl *via* acid displacement by nitric and sulfuric acid, particularly in relatively polluted marine air masses (Newberg et al., 2005; Sturges and Shaw, 1993). Whilst the air masses may be influenced by coal burning in China, they will also contain a vast mixture of other anthropogenic pollutants *e.g.* $NO_x$, $SO_x$, nitric acid and sulfuric acid. The presence of such pollutants means that as the air is transported over the South China Sea to the Bachok measurement site, there is potential for significant Cl⁻ depletion to occur. Lines 616 – 641 have been updated to explain this.

10. Line 522 – 523: It is hard to say that sulfate suppressed the formation of nitrate. The possible cause should be the deficiency of NH3, leading to the incomplete neutralization of sulfate and nitrate.

This section has been explained in more detail (lines 653-657). These measurements may be linked to each other through the important role of $H_2SO_4$ in the atmosphere. Acid displacement, when sea salt reacts with $H_2SO_4$, leads to the removal of Cl⁻ from the aerosol as gaseous HCl, and a partitioning of $SO_4^{2-}$ to the aerosol as $Na_2SO_4$. Furthermore, under an ammonia-poor regime (as observed in this study), $H_2SO_4$ has a lower vapour pressure than $HNO_3$, leading to the preferential formation of ammonium sulfate over ammonium nitrate when there is insufficient $NH_3$ available to fully neutralize sulfate and nitrate (Seinfeld and Pandis, 2006).

Minor Comments:

Page 6, Line 191: Taiwan is not a country.

This sentence has been adjusted accordingly (line 245).

Line 204: Ashfold et al. ; Line 222: Dominick et al.; Line 322: Oram et al.; Line 445: Freitas et al.; Line 456: Carlton et al.; Line 466: Huang et al.; The format is incorrect.

The format of the in-text citations has been corrected.

Line 335: what does "a NAME trajectory" mean?

The UK Met Office's Numerical Atmospheric Dispersion Modelling Environment (NAME) is used to model a range of atmospheric dispersion events. The full name has been provided in the text (line 424-426).

**Anonymous Referee #1**

The manuscript by Farren et al. entitled as 'Chemical Characterisation of Water-soluble Ions in Atmospheric Particulate Matter on the East Coast of Peninsular Malaysia' presents the observation data at Malaysia. The method and data quality seem to be reasonably good. The data in the manuscript could be a good addition to the existing data set in the region. Quality of figures and descriptions could significantly be improved. I provide some comments related to the presentation quality below. It would be good if the authors could significantly improve it.

Comments

L45 'During the northern hemisphere winter, a large anticyclone forms over Siberia each year, creating strong north-easterly monsoon winds in the South China Sea (Northeast Monsoon). These strong north-easterlies can transport air masses from rapidly developing East Asian countries (e.g. China, Japan, Taiwan, Vietnam, North and South Korea) across the South China Sea to the Maritime Continent.

' I am not sure if the statement is true. Please add references to support the description.

References have been added to support this statement (lines 48-49).

L185 'Figure 2 shows the 10-day backward air mass trajectories arriving at the measurement site during the demonstration campaign.'

Further details of the back-trajectory calculations, such as altitude, will be needed. It is not clear to me if a back trajectory analysis in the troposphere could provide a reliable result for such a long time-scale.

Further details of the back trajectories have been provided in the Supplement (Table S3). The back trajectories have been reduced to 7-day trajectories to better represent the troposphere. Figure 2 has been updated to colour the backward air mass trajectories by the altitude of the air mass.

L224 'and it is likely that the remainder was comprised primarily of organic aerosol'

Please provide a supporting information on this statement.

This sentence has been removed. It is not an important addition to the manuscript.

L240 'the mean $Na^+$/$Ca^{2+}$ ratio in the crust and mean $Ca^{2+}$/$Na^+$ ratio in seawater have been estimated as 1.78 w/w and 0.038 w/w respectively (Bowen, 1979)' I wonder how stable these values are. The uncertainties in the values directly influence the following discussion. Please provide a detailed description, rather than simply referring one publication.

$Na^+$ and $Ca^{2+}$ are both dominant cations in seawater with long residence times (1 million and 68 million years respectively). Although the total salt concentration or salinity of seawater varies somewhat, the ratio of the concentrations of major constituents to $Cl^-$ are remarkably constant and the ocean is well-mixed (Bowen, 1979). There may be more uncertainty in the mean $Ca^{2+}$/$Na^+$ crustal ratio, as it is more challenging to predict the elemental composition of the crust. The ratio used in this study is based on the assumption that the crust consists of 50% basalt and 50% granite (Taylor, 1964). Despite potential uncertainties, as reported by Becagli et al. (2005) and Boreddy and Kawamura (2015), this approach provides greater accuracy than simply using total $Na^+$ as a sea-spray marker. Importantly, none of the overall trends are drastically altered by using this approach. For example, similar correlations are observed for oxalate and $nss$$SO_4^{2-}$ (R = 0.69) as oxalate and total $SO_4^{2-}$ (R = 0.68) etc. Furthermore, ratios such as $Cl^-$/$ss$$Na^+$ are not hugely different to $Cl^-$/$total$$Na^+$ (average molar ratios are 0.40 and 0.36 respectively). This has been explained more carefully in section 3.3.1 (lines 299-301).

L314 'Air masses in cluster 10 passed over the megacity of Manila in the Philippines, but may have slightly lower $SO_4^{2-}$ levels due to the height of the back trajectories;'

I am unable to judge if the statement is valid, as no information about altitude is provided in the manuscript.

Please refer to the response to comment 5 (referee #2).

L351 'The uptake of $SO_4^{2-}$ is preferential to the uptake of $NO_3^-$ because sulfuric acid has a lower vapour pressure than nitric acid, and aqueous or solid $(NH_4)_2SO_4$ is the preferred form of sulfate'

The statement is unclear to me. Please clarify.

This statement has been clarified (lines 441-443). The average $NH_4^+$/$SO_4^{2-}$ molar ratio was 0.81, which indicated that there was insufficient gaseous $NH_3$ in the atmosphere to neutralise $SO_4^{2-}$. Under an

ammonia-poor regime, the uptake of $SO_4^{2-}$ is preferential to the uptake of $NO_3^-$ because sulfuric acid has a lower vapour pressure than nitric acid (Seinfeld and Pandis, 2006).

L402 'The ambient temperature and relative humidity data were taken from the measurements made nearby at the Sultan Ismail Petra airport.'

It seems to me that the authors assumed an internal mixing state in using the thermodynamic model. Is there any supporting evidence on this assumption?

One of the key assumptions of ISOROPPIA-II is that particles are internally mixed. This may limit the accuracy of the pH prediction, in additional to other limitations, such as the lack of gaseous $NH_3$ and $HNO_3$ measurements. However, assuming the particles are internally mixed is not unreasonable for this study. Firstly, there is evidence that aerosol arriving at the Bachok research station is often aged and hence tends to be internally mixed. Secondly, relative humidity remained high throughout the study (average = 77%). A short discussion of the internal mixing state assumption has been incorporated into section 2.6 (lines 207-213).

Figure 7

Almost all the trajectories look similar to me, except for C10. Please provide the detailed reasoning for classification.

Cluster analysis is used on back trajectories to group similar air mass origins together. Back trajectories with similar geographic origin and grouped together to gain information on pollutant species with similar chemical histories. A distance matrix is used to created a required number of clusters (*e.g.* n = 5) with the most different air mass trajectories. This has been explained in lines 364-366. As shown in an updated version of Figure 7 (Figure 8), the cluster analysis has been altered so that fewer clusters are now used (n = 5). These clusters are sufficiently different enough, in terms of their geographic origin and altitude, to describe the effect of different air mass origins on pollutant species.

Minor comments

L76 'Dominick et al. characterised ..' I believe that it should be written as 'Dominick et al. (2015) characterized. . .' There are many similar descriptions when the authors cite other publications. Please check the recent publications of the journal carefully in preparing a manuscript.

The format of the in-text citations has been corrected throughout the text.

L200 'The station is located approximately 23 km away at the Sultan Ismail Petra airport in Kota Bharu (6.17298N, 102.2928E), as shown in Fig. S1 (Supplement)'

A similar information has already appeared at L150. Please minimize duplicated descriptions.

The duplication of this description has been minimized accordingly.

**References**

Becagli, S., Proposito, M., Benassai, S., Gragnani, R., Magand, O., Traversi, R., and Udisti, R.: Spatial distribution of biogenic sulphur compounds (MSA, nssSO(4)(2-)) in the northern Victoria Land-Dome C-Wilkes Land, area, East Antarctica, Ann Glaciol, 41, 23-31, Doi 10.3189/172756405781813384, 2005.
Boreddy, S. K. R., and Kawamura, K.: A 12-year observation of water-soluble ions in TSP aerosols collected at a remote marine location in the western North Pacific: an outflow region of Asian dust, Atmos Chem Phys, 15, 6437-6453, 10.5194/acp-15-6437-2015, 2015.
Bowen, H. J. M.: Environmental chemistry of the elements, Academic Press, London, 1979.

Newberg, J. T., Matthew, B. M., and Anastasio, C.: Chloride and bromide depletions in sea-salt particles over the northeastern Pacific Ocean, J Geophys Res-Atmos, 110, Artn D06209

10.1029/2004jd005446, 2005.

Seinfeld, J. H., and Pandis, S. N.: Atmospheric chemistry and physics : from air pollution to climate change, 2nd ed., J. Wiley, Hoboken, N.J., 1-1203 pp., 2006.

Sturges, W. T., and Shaw, G. E.: Halogens in Aerosols in Central Alaska, Atmos Environ a-Gen, 27, 2969-2977, Doi 10.1016/0960-1686(93)90329-W, 1993.

Taylor, S. R.: Abundance of Chemical Elements in the Continental Crust - a New Table, Geochim Cosmochim Ac, 28, 1273-1285, Doi 10.1016/0016-7037(64)90129-2, 1964.

---

## Author Response (AR2)

**Response to referees: Chemical Characterisation of Water-soluble Ions in Atmospheric Particulate Matter on the East Coast of Peninsular Malaysia**

The authors addressed part of the comments while some critical comments were not satisfactorily responded or fully explained.

As for comment #4, the authors replied that "the levels of K+ do not rule out the influence of biomass burning on a regional/national scale." At least, the authors should check whether there were biomass burning events on a regional/national scale by looking at fire hotspots from satellites. Secondly, the authors responded that "Biomass burning aerosol in the wider region will have undergone atmospheric processing and dispersion prior to arrival at the measurement site, potentially lowering
K+ levels and raising the concentration of secondary species." If this statement is correct, sulfate should also be subject to the similar process as the authors found sulfate derived from long-range transport. However, the concentrations of sulfate maintained at high levels and I didn't see an obvious dispersion effect.
There is one paper published in Scientific Reports that showed that the signal of biomass burning could be detectable from the north pole to the south pole from cruise measurement. The concentrations of K+ less than 1ug/m3 were likely representative
of the background signal of biomass burning emissions over the region but didn't necessarily point to the biomass burning activities in a wider region.

The authors have taken the reviewers advice and investigated the occurrence of fires in the region surrounding the site using data obtained from the moderate-resolution imaging spectroradiometer (MODIS) instrument on board the NASA Terra satellite
(Giglio et al. 2003). Fire maps were accessed *via* the global forest watch website (GFW, 2018) and are shown in Figure S4 for the measurement period discussed in this paper. It is clear that there is widespread biomass burning throughout SE Asia during our measurements. Lines 562-565 have been amended to explain this point more clearly. However, the most important point of this discussion in the manuscript is the observed correlation of nssK$^+$ with the oxalate ion, and the nssK$^+$/oxalate ratio. Whilst K$^+$ levels may be relatively low (less than 1 μg m$^{-3}$), there is a strong positive correlation between nssK$^+$ and oxalate,
and a low nssK$^+$/oxalate ratio. This indicates that nssK$^+$ from biomass burning (which has been observed in the wider region) may be an important source of secondary oxalate, rather than a primary source (lines 564 – 572).

Comment #5: The authors explained the high concentrations of sulfate in Cluster 5 came from higher altitudes in a low pressure system (no more explanations were offered). Then in the response to Comment #9, the authors explained that the concentrations
of Cl- were low, which were also due to the air masses at high altitudes. Those inconsistent analysis and discussions in different sections of the manuscript are common in this study.

This point has been explained more clearly (lines 416 - 425). For clusters 1 – 4, the highest $SO_4^{2-}$ concentrations are seen when air masses arriving at the site come from low altitudes and have significant continental influence from industrialised regions of East Asia. The lowest $SO_4^{2-}$ concentrations are seen for marine air masses that have passed over the South China Sea.

Cluster 5 shows very different behaviour. We disagree that we have been inconsistent in our discussion and have presented the data clearly. The air masses in cluster 5 arriving at the site are not a result of north-easterly winds from East Asian countries travelling across the South China Sea. Instead, there is evidence of a cyclonic weather system off the coast of the island of Borneo over the South China Sea. The air masses in cluster 5 arrived during the final 24-hour period of the measurement campaign and only one $SO_4^{2-}$ measurement is available (18.1 ug m$^{-3}$). It is not possible to fully understand the nature of air masses from this different region without further measurements.

Figure 6 & Figure 7,8

The pollution rose plot (Figure 6) of SO2 shows that there were almost no prevailing winds from the north and high concentrations of SO2 mainly came from the west and south. But the backward trajectories analysis (Figures 7,8) dominantly derived from the north. I'm really confused that how could the analysis of meteorological conditions differed so much.

The author has misunderstood the data that is presented in Figure 6 (now Figure 5). As explained in lines 362 – 372, the pollution rose plot of SO2 (Fig 6) only showed $SO_2$ concentrations above 5 ppb *i.e.* the elevated $SO_2$ events. The pollution rose excluded $SO_2$ concentrations below 5 ppb.

The plot showed that most of the spikes in $SO_2$ were observed in calmer conditions when the air arriving at the site had passed over nearby land to the south west of Bachok.

A direct comparison with wind directions presented in figures 7 and 8 is not appropriate because figures 7 and 8 represent the back trajectories across the entire measurement period, rather than solely during the high $SO_2$ events.

The trajectories focus on long-range transport and represent the dominant overall meteorological conditions, which were strong north-easterlies from East Asia and the South China Sea. The dominance of north-easterlies has been further illustrated in Figure S2 in the Supplement, which shows hourly wind rose plots averaged across the measurement campaign.

An additional pollution rose has been added to Figure 6 to make this point clear. The left panel represents recorded $SO_2$ concentrations $\geq$ 5 ppb and the right panel represents all $SO_2$ data recorded during the measurement period.

**Frequency of counts by wind direction (%)** **Frequency of counts by wind direction (%)**

Overall, I didn't see great improvement of the presentation quality of the manuscript. Both referees requested improving the quality of figures and descriptions while the authors didn't take the suggestions. Data analysis is presented in a very simple manner and a lot of the discussions are based on hypothesis without firm evidence or other supportive data. Most importantly, contradictions were often found which makes the manuscript unreadable. I personally don't think the current version of this manuscript meets the quality of the journal.

The quality of figures and descriptions has been significantly improved throughout the entire manuscript. The authors has improved the quality of discussions where necessary, and clarified the misunderstandings surrounding Figures 6, 7 and 8. The author believes that this version of the manuscript meets the quality of ACP.

[revised manuscript text omitted]

---

## Author Response (AR3)

**Response to editor: Chemical Characterisation of Water-soluble Ions in Atmospheric Particulate Matter on the East Coast of Peninsular Malaysia**

The revised manuscript has addressed most concerns of the referee. For the following question "Sulfate should also be subject to the similar process as the authors found sulfate derived from long-range transport. However, the concentrations of sulfate maintained at high levels and I didn't see an obvious dispersion effect", please explain why high concentrations of sulfate were still observed despite the dispersion effect.

In the manuscript, we presented the measured water-soluble ions observed in aerosol at the Bachok research station. Potassium is a primary pollutant and may be subject to different loss processes to sulfate, which is a secondary pollutant and may also form during transport. Whilst $K^+$ is directly emitted e.g. from biomass burning, $SO_4^{2-}$ can form via several $SO_2$ oxidation pathways and is also likely to come from a variety of sources. Therefore, it is possible that high concentrations of sulfate can be present at the Bachok research station.

[revised manuscript text omitted]